# FROM WORDS TO WIRES: Generating Functioning Electronic Devices from Natural Language Descriptions

**Peter Jansen**
University of Arizona
pajansen@arizona.edu

## Abstract

In this work, we show that contemporary language models have a previously unknown skill – the capacity for electronic circuit design from high-level textual descriptions, akin to code generation. We introduce two benchmarks: PINS100, assessing model knowledge of electrical components, and MICRO25, evaluating a model's capability to design common microcontroller circuits and code in the ARDUINO ecosystem that involve input, output, sensors, motors, protocols, and logic – with models such as GPT-4 and Claude-V1 achieving between 60% to 96% PASS@1 on generating full devices. We include six case studies of using language models as a design assistant for moderately complex devices, such as a *radiation-powered random number generator*, an *emoji keyboard*, a *visible spectrometer*, and several *assistive devices*, while offering a qualitative analysis performance, outlining evaluation challenges, and suggesting areas of development to improve complex circuit design and practical utility. With this work, we aim to spur research at the juncture of natural language processing and electronic design.[1][2]

## 1 Introduction

The realm of science fiction often presents us with captivating visions of technology's future. A case in point is the *replicator* from the TV series *Star Trek*, a machine capable of creating various physical objects – from food and medicine to functioning devices – based solely on a user's high-level description of those objects. Contemporary language models hint at the precursors to some of this capacity for design, including the ability to design novel 2D and 3D object models (Ramesh et al., 2022; Nichol et al., 2022), predict molecular structures for drug discovery (Liu et al., 2021;

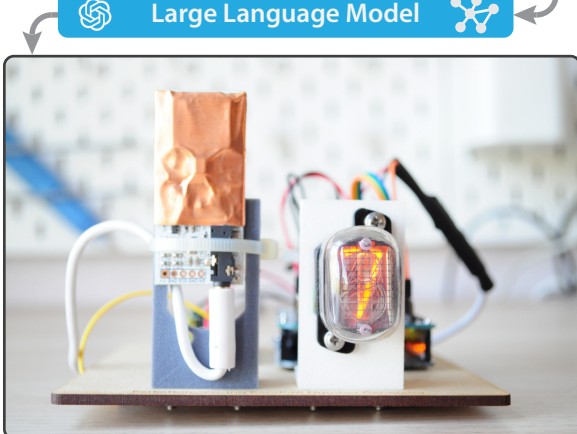

"Design a random number generator that uses ambient radiation level to set the random seed, and displays random numbers on a nixie tube."

Large Language Model

Figure 1: An example of using a language model to convert a high-level textual description of an electronic device into the designs for that device. Those designs are then reviewed by a domain expert, and any errors corrected, before the device is physically manufactured using rapid-prototyping techniques.

Flam-Shepherd and Aspuru-Guzik, 2023), and generate increasingly complex sections of code that power a variety of user applications (Li et al., 2023; Wang et al., 2023).

In this work, we show that language models have a previously unknown skill – the capacity to generate working electronic devices from high-level textual descriptions – effectively bridging the gap between the *words* of a device description to the *wires* of a device design. The design process for electronic devices, such as the random number generator in Figure 1, typically follows a stage-like process illustrated in Figure 2. These steps include: ideation, electronic design (including generating parts lists, electronic schematics, and code for embedded processors called *microcontrollers*), followed ultimately by physical implementation

---

[1] PYTHON library and data: https://github.com/cognitiveailab/words2wires

[2] Companion video: https://youtu.be/PZ1rr0dDAPI

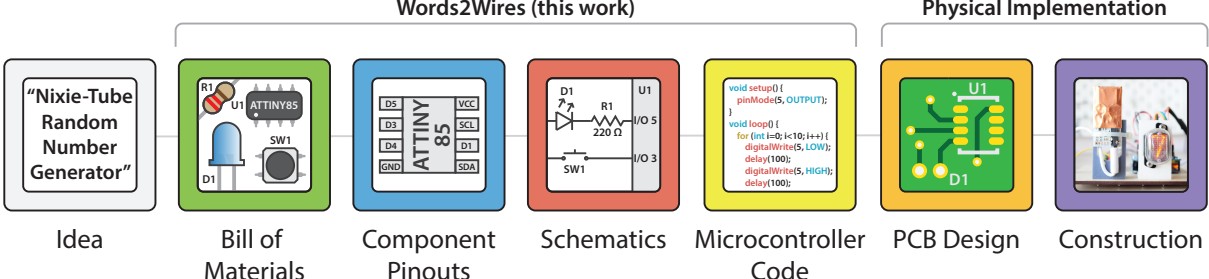

Figure 2: An overview of the electronics device design process, from concept to design implementation.

via manufacturing. Our focus in this work is on automating the task of transforming a high-level concept into a practical electronics schematic, complete with companion microcontroller code—a task that currently requires significant human expertise and effort. We evaluate our approach by constructing these devices either in simulators or, for six open-ended case studies, as physical devices implemented using rapid prototyping techniques such as breadboards, circuit board manufacture, laser cutting, and 3D printing.

The contributions in this work are:

1. We empirically demonstrate the novel capacity for language models to design electronic devices from high-level textual descriptions, and introduce two benchmarks to measure this capacity. The first, PINS100, measures knowledge of 100 common electronic components. The second, MICRO25, measures the ability to design 25 common electronic devices from start to finish, including the generation of *bills-of-materials*, *pinouts*, *schematics*, and *code*.

2. Our experimental evaluation shows that both GPT-4 and CLAUDE-V1 have moderate-to-strong performance on these benchmarks, with GPT-4 achieving 96% PASS@1 at generating correct schematics and functioning code on MICRO25, while CLAUDE-V1 scores 60% on schematics and 76% on code.

3. We present six real-world case studies of using language models to construct novel devices, including an *emoji keyboard*, a *visible light spectrometer*, and two *assistive devices*. Alongside, we provide a qualitative assessment of the strengths of current language models for electronic device design, while also outlining the challenges in enhancing performance, automating evaluation, and increasing their practical utility.

## 2 Related Work

**Arduino microcontroller ecosystem:** This work focuses on circuits that are controlled by *microcontrollers*, which are processors intended for embedded applications. In particular, we focus on microcontrollers supported by the ARDUINO ecosystem (Banzi and Shiloh, 2022), a popular cross-platform set of C++ tools and libraries[3] intended to promote learning and lower the barrier to entry for creating physical devices. Most of the devices described in this work target the ARDUINO UNO, a platform with more than 10 million units sold as of 2023, which uses an ATMEGA328P microcontroller with 2K of RAM, 32K of program space, 20 input/output (I/O) pins, and a speed of 16MHz. While these specifications (e.g. RAM and speed) are modest for desktop computers, they are typical of microcontrollers and embedded applications where programs are typically run bare-metal (i.e. without an operating system).

**Code Generation:** Though aspects of physical design (such as circuit board layout) have been automated for decades (see Huang et al. (2021) for review), to the best of our knowledge, this is the first work to use language models for automating early-stage circuit design, converting high-level textual device descriptions into initial electronic schematics and microcontroller code. Designing code-driven electronic circuits is similar to code generation tasks (e.g. Chen et al., 2021; Austin et al., 2021; Hendrycks et al., 2021; Lai et al., 2022; Nijkamp et al., 2023, *inter alia*), with the additional requirement that a model must jointly generate a corresponding *electrical circuit schematic* that is compatible with the code, that together allow the microcontroller to accomplish a given task.

**Conditioned Code Generation:** Generating code from intermediate planning representations (such

---

[3] https://www.arduino.cc/

Figure 3: An example of representing a device specification as text (in formatted JSON) such that it can be generated by a language model, for a trivial device that illuminates an LED in response to a button being pressed. The device specification includes the bill of materials, component pinouts, schematic (represented as a netlist), and microcontroller code. The specification can then be used to create that device, either in simulation or through physical construction. The device shown here was generated by GPT-4, and edited slightly for space.

as UML diagrams) can increase task performance (Liu et al., 2022b), and here we adapt this to the electronics context by generating device specifications (such as schematics) immediately before code generation, to condition code generation on a specific device. Similarly, generating structured representations (in the form of code) can better elicit the knowledge in a language model (Madaan et al., 2022), where here we adapt this to generating device specifications as highly-structured JSON representations. These device specifications (bills of materials, pinouts, schematics, and code) are thousands of tokens long, and their generation is enabled by the increase in model context lengths to

8K tokens, allowing models to generate hundreds of lines of code (OpenAI, 2023; Li et al., 2023) compared to earlier models with smaller generation capacity (Chen et al., 2021; Li et al., 2022; Fried et al., 2023).

**Hardware Description Languages:** A contemporaneous body of work describes code generation tasks for hardware description languages (such as VHDL or VERILOG) that run on specialized processors called Field-Programmable Gate Arrays (FPGAS) – typically to prototype specialized CPU designs, or accelerate digital signal processing. Thakur et al. (2023) introduce a benchmark of 17 simple bitwise tasks, with the most advanced

tasks including bitwise addition or counting, and show the best performing model (CODEGEN-6B) to achieve 60% PASS@10 when solving these tasks. Similarly, Blocklove et al. (2023) use GPT-4 as a Verilog coding assistant, demonstrating that it can perform well at generating 8 simple Verilog designs (such a bitwise adder, or 3-state finite state machine), while also providing a qualitative evaluation of using GPT-4 as an assistant to design an 8-bit accumulator-based microprocessor. In contrast, where these projects generate code that runs on FPGAs, this work (WORDS2WIRES) generates both *electrical schematics* and code for electrical devices that are built from – and interface with – real electrical components, such as sensors, motors, and displays, while also providing a larger-scale set of benchmarks that perform significantly more complex and real-world tasks.

**Single-shot vs Collaborative:** In this work, we investigate device generation in two contexts. First, we assess single-turn generation of error-free devices using the MICRO25 benchmark, a set of electronic design tasks evaluated using code metrics similar to PASS@1 (Kulal et al., 2019) that require generating a single correct solution, with strict binary measures of task success. In the second context, we explore a collaborative coding-assistant setting, where prompts can be iteratively refined, and any errors in the schematics or code corrected by the end user. This approach is akin to tools such as GITHUB COPILOT (Chen et al., 2021), which assist in generating short parts of programs. However, in our case, the model is utilized to generate an initial version of the entire project, after which the user corrects any errors before physically constructing the device.

## 3 Experiment 1: Component Knowledge

To design electronic circuits, the designer needs a knowledge of the individual electrical components that can be used to build a circuit. One of the most fundamental aspects of this knowledge is the *component pinouts*, or the specific function of each electrical terminal (or *pin*) on a component. For example, a light-emitting diode (LED) typically has two pins, one an *anode* where positive voltage is applied, and one a *cathode* where the negative terminal or ground is applied. Here we measure large language models' knowledge of component pinouts by asking them to generate pinouts for a large number of common electrical components.

| Pinout Scoring Method | GPT3.5 | GPT-4 | Claude |
|---|---|---|---|
| Strict Scoring (Exact) | 55% | **74%** | 56% |
| Permissive Scoring | 74% | **86%** | 72% |

Table 1: Model accuracy in generating accurate pinout information for 100 common electronic components. *Strict scoring* requires all generated pins on a given device to be accurate in order to be counted as correct. *Permissive scoring* requires pins critical to the function of a given component to be correct, but still counts generations with non-critical missing pins (such as when the device is mounted to a breakout board) to be correct.

**Component Pinouts:** While LEDs and other common components such as resistors and capacitors commonly have two pins, other components have varying numbers of pins, each with different functions. For example, a common kind of digital motor called a *hobby servo* typically has three pins: an anode, a cathode, and a digital signal pin that accepts a pulse from a microcontroller that signifies what angle the motor should turn to. Common sensors generally have between two and ten pins, depending on the communication protocols (e.g. digital vs analog) they employ. Similarly, integrated circuits, and microcontrollers in specific, can have as few as 8 pins (such as the ATTINY85 microcontroller in Figure 2), but frequently have dozens and occasionally hundreds of unique pins. Connecting component pins incorrectly in a schematic will cause a device to malfunction, so this knowledge is critical to constructing working circuits.

**Benchmark:** We assembled a benchmark of electronic component pinouts, PINS100, containing 100 common parts frequently used in circuits found on high-traffic electronic tutorial websites such as the ARDUINO PROJECT HUB and AUTODESK TINKERCAD CIRCUITS. Components range from 2 to 40 pins, and span a large assortment of part categories including passives (e.g. *resistors/capacitors*), input (e.g. *switches*), output (e.g. *LEDs, motors, relays*), sensors, integrated circuits, power regulators, logic (e.g. 7400-SERIES *AND and OR gates*), and microcontrollers (e.g ARDUINO, RASPBERRY PI).

**Models:** We evaluate on instruction-tuned models including OpenAI's CHATGPT (GPT-3.5-TURBO) and GPT-4 (OpenAI, 2023), and Anthropic CLAUDE-V1[4]. Model prompts are identi-

---

[4] https://www.anthropic.com/

cal across models, and include a static 1-shot exemplar (a 14-pin 7400-SERIES logic integrated circuit) that provides an example of the pinout task, as well as the requested JSON output format. Additional hyperparameters, evaluation details, and the full prompt are provided in the APPENDIX.

**Evaluation:** We evaluate using two binary measures of accuracy analogous to the PASS@1 code-generation metric (Kulal et al., 2019). The first scoring method, *strict*, requires a given model to output all of a component's pins correctly to be considered correct, otherwise it will be considered incorrect. The second method, *permissive*, requires only the function-critical pins of a component to be present to be considered correct, while failing to include non-critical pins still counts as success.[5]

**Results:** Model performance in the pinout generation task is shown in Table 1. Performance reflects average binary PASS@1 performance of a given model on generating accurate pinouts – for example, a score of 50% reflects that 50% of the components had completely correct pinouts. Here, GPT-4 achieves the highest *strict* scoring performance, generating accurate pinouts for 74% of components, while both GPT3.5 and CLAUDE-V1 achieve similar levels of performance, generating correct pinouts for 55% and 56% of components, respectively. *Permissive* scoring increases performance, with the best-scoring GPT-4 model capable of generating pinouts that include the most critical pins for 86% of electrical components in the benchmark. Taken together, these results suggest that large language models have a moderate-to-strong knowledge of electrical component pinouts, a core requirement for designing functioning electronic circuits.

## 4 Experiment 2: Circuit Generation

How well can contemporary language models leverage their component knowledge to design simple but functioning electronic devices? In this experiment, we investigate end-to-end generation of working devices, which includes generating four core elements: (1) a *bill of materials (BOM)*, or list of components in the device, (2) the *pinouts* for each component, (3) a complete electrical circuit diagram called a *schematic* that details how the components are to be connected, and (4) the *code*

to be programmed onto a microcontroller – a lightweight processor that controls embedded circuits. An example of a complete model-generated design for a trivial device that turns on an LED in response to a button being pressed is shown in Figure 3.

**Benchmark:** To assess a model's ability to create microcontroller-driven electronic devices, we developed a benchmark, MICRO25, that includes 25 tasks intended for the common ARDUINO microcontroller ecosystem.. These tasks, shown in Table 2, span 5 core categories including: input, interface protocols, output, sensors, and logic. Each task is either tailored to test a specific fundamental competency required to build basic microcontroller-driven electronic devices common in introductory microcontroller curricula, or the integration of several competencies into larger design flows.[6]

**Representations:** Models were given format prompts to export all generated elements (*bill of materials*, *pinouts*, *schematics*, and *code*) in an annotated JSON format, shown in Figure 3. The annotated format allows the model to add comments for each generated element (e.g. specifying the uses of each component, or the purpose of each connection in the schematic), analogous to chain-of-thought reasoning (Wei et al., 2022) applied to circuit generation, as well as code generation from requirements specifications (Liu et al., 2022b), where here the requirements are the schematics and other device specifications generated immediately preceding the code. Additional details of this representation format are provided in APPENDIX B.4 and C.

**Models:** We evaluate on instruction-tuned models with large (8k token) context windows, including OpenAI's GPT-4 (OpenAI, 2023) and Anthropic CLAUDE-V1. Prompts are identical across both models, and include a static minimal 1-shot example of generating each of the 4 elements of a device specification (*bill of materials*, *pinouts*, *schematic*, *code*) in the desired JSON output format. In response to specific types of errors identified during pilot studies, the prompt also includes three incomplete snippets that provide portions of two positive and one negative generation example. After initial generation, the models are given a reflection

---

[5]Additional details of permissive scoring (including an example) can be found in APPENDIX B.1.1.

[6]MICRO25 and PINS100 require partially-overlapping component knowledge. Solving MICRO25 requires some (but not all) of the components in PINS100. Similarly, each MICRO25 task has many possible solutions, and with hundreds of thousands of different electrical components currently available, it is possible to generate solutions whose schematics use one or more components not listed in PINS100.

| Category | Task Description | Schematic (GPT-4) | Code (GPT-4) | Schematic (Claude) | Code (Claude) |
|---|---|:---:|:---:|:---:|:---:|
| **Input** | | | | | |
| Digital - Button | Turn on an I/O pin when a single button is pressed. | ✓ | ✓ | ✓ | ✗ |
| Digital - Multiple | Turn on an I/O pin when exactly 2 of 4 buttons are pressed. | ✓ | ✓ | ✓ | ✗ |
| Analog | Read potentiometer, activate I/O pin when voltage exceeds 2.5V | ✓ | ✓ | ✗ | ✓ |
| **Protocols** | | | | | |
| Serial/UART | Read the serial port. When "hello" is read, respond with "world". | ✓ | ✓ | ✓ | ✓ |
| Serial/SPI | Connect using SPI. When "hello" is read, respond with "world". | ✓ | ✓ | ✓ | ✓ |
| I2C | Read a value from a specific I2C address and register every second. | ✓ | ✓ | ✓ | ✓ |
| SD Card | Open a file, randomly append one of 4 strings every 10 seconds. | ✓ | ✓ | ✓ | ✓ |
| **Output** | | | | | |
| Motor - DC | Rotate a DC motor clockwise then counterclockwise every 5 seconds. | ✗ | ✓ | ✗ | ✓ |
| Motor - Stepper | Oscillate a stepper motor 45 degrees every 5 seconds. | ✓ | ✓ | ✗ | ✗ |
| Motor - Servo | Continuously move a servo back and forth from 45 to 135 degrees. | ✓ | ✓ | ✓ | ✓ |
| LED - Blink | Blink an LED every 500 milliseconds. | ✓ | ✓ | ✓ | ✓ |
| LED - Sequence | Blink 4 LEDs in sequence, once every 500 milliseconds. | ✓ | ✓ | ✗ | ✓ |
| LED - 7 Segment | Count from 0 to 9 on a 7-segment display, changing every 500 msec. | ✓ | ✓ | ✓ | ✓ |
| LED - Neopixel | Slowly move through a rainbow of colours on an RGB LED. | ✓ | ✓ | ✓ | ✓ |
| Relay | Turn on a relay for 2 seconds, then off for 5 seconds. | ✓ | ✓ | ✗ | ✓ |
| LCD | Print "Hello World" on a 16x2 LCD (HF44780 compatible controller) | ✓ | ✓ | ✗ | ✓ |
| Sound - Buzzer | Cycle between high, medium, then low tones on a Piezo Buzzer. | ✓ | ✓ | ✗ | ✓ |
| Analog Output | Produce a sawtooth wave, ramping up from 0V to 5V every 50 msec. | ✓ | ✓ | ✗ | ✓ |
| **Sensors** | | | | | |
| Resistive - CDS | Read the light intensity from a resistive sensor (CDS cell). | ✓ | ✓ | ✗ | ✓ |
| Manual Protocol | Read the distance from a HC-SR04 ultrasonic distance sensor. | ✓ | ✓ | ✓ | ✓ |
| I2C - Magnetic | Read x/y/z components of magnetic field using an I2C magnetometer. | ✓ | ✗ | ✓ | ✓ |
| **Logic** | | | | | |
| Simon | Create the popular memory game Simon, using 4 colors. | ✓ | ✓ | ✓ | ✗ |
| Conway | Create the popular Conway's Game of Life, on an 8x8 LED matrix. | ✓ | ✓ | ✗ | ✗ |
| Clock | Create a clock on a 16x2 I2C LCD, with buttons for setting the time. | ✓ | ✓ | ✓ | ✗ |
| Air Temperature | Read the air temperature, display temperature on an RGB LED. | ✓ | ✓ | ✓ | ✓ |
| **Overall Performance** | | **96%** | **96%** | **60%** | **76%** |

Table 2: Model performance (PASS@1) on the MICRO25 benchmark generation tasks, broken down by *schematic* and *code*. Task descriptions are summarized for space, where full task descriptions can be found in APPENDIX E.

prompt containing 12 common errors (such as correctly supplying power to each component, explicitly enumerating each connection in the schematic, and having code that functions as intended), and allowed to iteratively reflect and improve output until providing a specific stop token signifying that the model has detected no further errors. The initial prompt requires 1884 tokens, and the reflection prompt requires 431 tokens. Additional model details, including evaluation details and the full prompt are provided in the APPENDIX.

**Evaluation:** Generated devices are broken down into electrical (schematic) and code components, each of which is separately evaluated using a binary PASS@1 metric (i.e. functional or non-functional).

Because the schematic subsumes the *bill of materials* and *component pinout* information, we do not evaluate the BOM or pinouts independently, only the entire schematic – but do still include generating the BOM and pinouts in the prompt to facilitate chain-of-thought reasoning. Due to the challenges of automatic evaluation in this domain, evaluation was conducted manually by a domain expert through inspection, simulation, and physical circuit construction.

**Results:** The results of the device generation task on the MICRO25 benchmark are shown in Table 2. Across all tasks, model-generated code ranged from a minimum of 13 lines to a maximum of 145 lines (average 38 lines per program). GPT-4 per-

forms extremely well on the MICRO25 benchmark, correctly generating schematics and code for 96% of benchmark tasks. CLAUDE-V1 exhibits more modest performance, achieving 60% for schematic generation, and 76% for microcontroller code generation. Taken together, this shows that contemporary language models have moderate-to-excellent overall capacity for generating common electrical circuits end-to-end, from bills of materials, pinouts, and schematics, to paired microcontroller code that accomplishes the desired functionality.

## 5 Experiment 3: Open Device Generation

While we've observed that language models have the capacity to design comparatively simple devices in Experiment 2, this result is tempered by these benchmark tasks being representative of fairly common skills and capacities that are frequently taught in microcontroller-oriented curricula found in books, internet tutorials, and blog posts – and as such, the MICRO25 tasks likely exist in some form in the voluminous (but closed) training data of these models. Here, we examine how well the best-performing model, GPT-4, can create comparatively more complex devices in a more realistic and qualitative setting, where it is used as a *design assistant* like GITHUB COPILOT (Chen et al., 2021) to create initial plans that are then vetted and corrected by a domain expert before being physically constructed. To further increase task difficulty, each of the device specifications in this experiment were explicitly crafted to be highly unusual – either using uncommon components, or combining common components in unusual ways – such that the likelihood of similar devices appearing in the closed model training data is low. The six devices are shown in Figure 4.

**Methods:** Initial specifications (in the form of a natural language textual description of a device) were created for all devices, and iteratively refined several times to provide clarifications in response to undesired or errorful model-generated output. After several attempts at refining device descriptions, any remaining errors were manually corrected by a domain expert, then the devices were physically manufactured. All devices were constructed by a human, and physical aspects of the design (e.g. printed circuit boards, 3D printed or laser cut components) were designed by a human. Full device description strings, model prompts, and detailed qualitative descriptions of manual corrections required by a domain expert to reach functionality are described in the APPENDIX.

**Qualitative Challenges:** Generation challenges can be organized into two categories: prompt-specific challenges, and hardware-specific challenges. With respect to prompts, device generation is *highly sensitive* to the specific prompt, and small (and seemingly helpful) changes in the prompt to address an error can cause new errors to occur in other aspects of the device that were previously generated correctly. Similarly, having many composite requirements in the project (such as adding the requirement for each key in the *emoji keyboard* to generate its own musical tune) generally decreases performance, and suggests that iteratively generating devices from the core requirements (such as generating a functioning emoji keyboard) through to more fine-grained details (like adding in musical tunes) may reduce the inference load at each generation step, improving generation performance.

With respect to hardware, a number of pragmatic issues occur. Electronic parts regularly reach end-of-life cycles and are no longer manufactured or easily available, yet the model frequently generates these, likely due to the abundance of examples that use these components in internet tutorials. Similarly, the model frequently uses deprecated versions of libraries, or combines the features from different versions of libraries. Finally, the model generally performs poorly at generating low-level device drivers for specific hardware (such as sensors), and favors using existing device driver libraries. When an existing library isn't available, the model will either hallucinate one, or generate a reasonable first-pass at a device library that requires extensive modification to low-level details (like clock timings) to function.

## 6 Challenges and Discussion

We identify the following challenges and opportunities in developing this capacity for automated device design further:

**Prompt Sensitivity:** Currently, small changes in the prompt can cause large changes in the output, affecting overall performance. While this is evident from small changes in task description strings producing novel errors in Experiment 3, this phenomenon is also visible for simpler cases. For example, the pinouts for some components – such as the MLX90614 temperature sensor – are incorrect when tested independently in Experiment 1, but

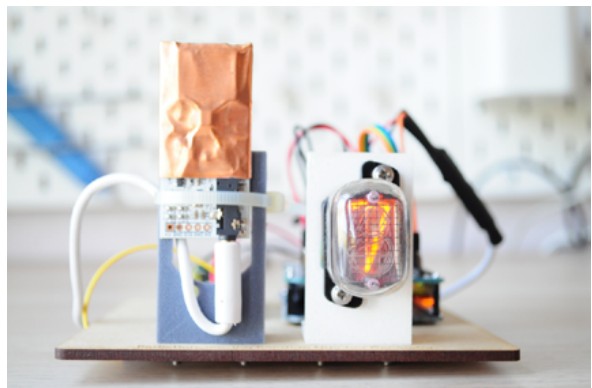

**Random number generator** that uses an ambient radiation sensor to continuously update the random seed. Random numbers are generated every few seconds, and displayed on a vintage nixie tube using a high-voltage driver.

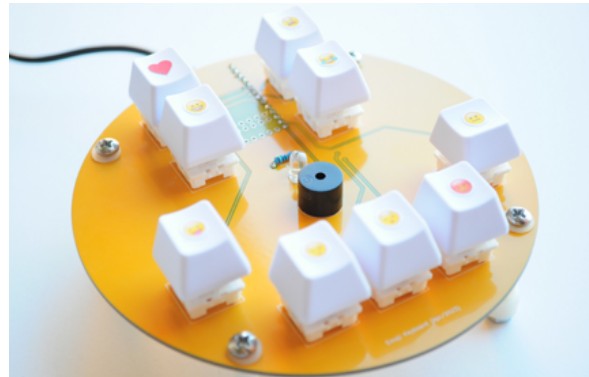

**Emoji USB keyboard** that has keys for 9 common emojis. Pressing an emoji types its ASCII string, just as if entered on a normal keyboard. A short musical tune with similiar affect to the emoji (e.g. a love song for the heart emoji) is also played.

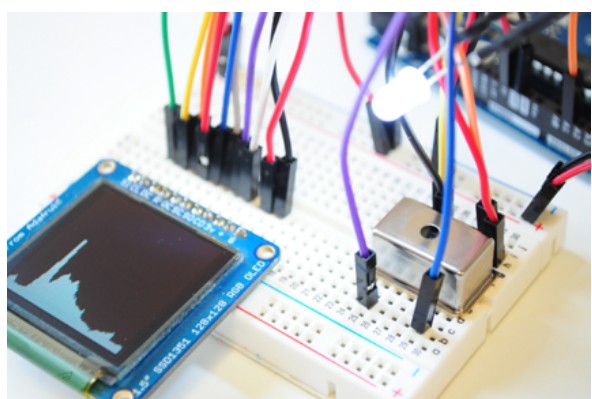

**Visible light spectrometer** that measures spectra using a Hamamatsu micro-spectrometer and displays the spectrum on a 128x128 pixel OLED screen. Here, the device is shown measuring the characteristic spectrum of a white LED.

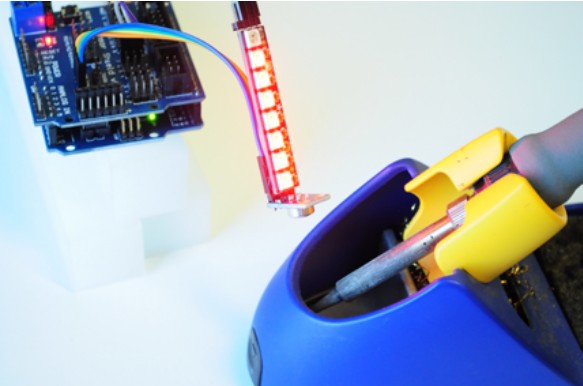

**Non-contact temperature sensor** that displays the temperature on a strip of 8 LEDs. Higher temperatures show as red colors and illuminate more LEDs, while lower temperatures show as blue colors and illuminate fewer LEDs.

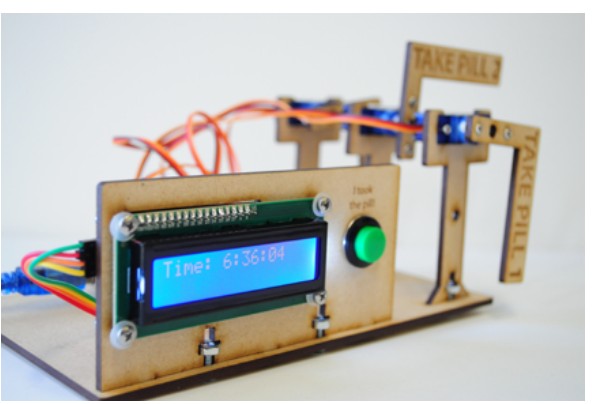

**Pill alarm assistive device** that has alarms for three pills. When it's time to take a pill, the alarm continuously waives a physical flag saying "Take Pill *X*" back and forth using a servo motor to get the users attention, until the button is pressed.

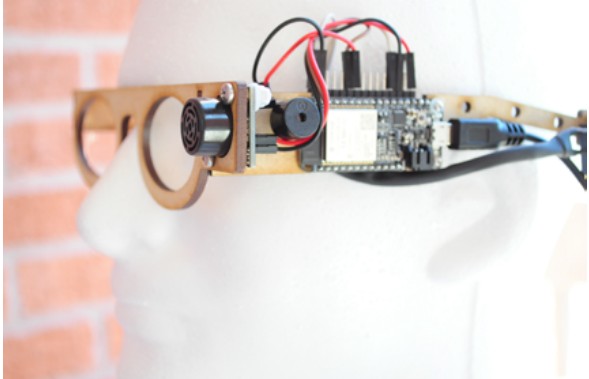

**Ultrasonic glasses assistive device** that uses an ultrasonic distance sensor to measure the distance of the nearest object in front of a person with a visual impairment. The distance is converted into an audible tone of varying frequency.

Figure 4: Six devices designed using WORDS2WIRES in the open generation condition, then physically constructed.

correct when generated as part of a full circuit in Experiment 2. This suggests that task performance is not currently robust, and may benefit from creating and fine-tuning on a task-specific dataset.

**Manual Evaluation:** Like COPILOT, we have observed that the electronic designs generated by language models are rarely perfect and frequently have errors. Currently these have to be discovered and corrected by a human. Contemporary work in code generation aims to use reflection (e.g. Shinn et al., 2023) to iteratively run generated code in an external interpreter (like PYTHON), report any errors

to the language model, then continue this process until the generated code runs error-free. The lack of electronic simulators with large libraries of simulated devices presents a significant barrier to this form of automatic evaluation in the near-term. Ultimately this may be addressed by constraining circuit generation to only parts available within a given simulator, or pushing a focused effort to developing more capable simulators with a larger repertoire of components.

**Generating devices with common-sense knowledge:** Language models contain a variety of common-sense reasoning abilities (West et al., 2022; Liu et al., 2022a), and leveraging these abilities may enable new applications. For example, in the context of assistive devices, GPT-4 is able to infer that an ultrasonic sensor can be used to create assistive glasses to aid the visually impaired with navigation. Similarly, the model can use its common-sense knowledge to design devices that contain the most common *emojis*, or keys for all the *prime numbers* up to 20 just as easily. Ultimately, electronic devices may be distributed as templates, that can be semi-automatically customized to a variety of applications based on user preferences.

**Quantifying time savings of automatic versus human device design:** Precisely quantifying the benefits of automated coding assistants (such as GITHUB COPILOT) is challenging, and currently measured at least in-part with qualitative measures (Ziegler et al., 2022). These assistants may provide large time savings when they function correctly, but likely increase debugging time when they generate problematic code, complicating measuring their precise benefit. A similar situation likely exists here for device generation, and we provide only the following anecdotal account: the best-performing GPT-4 model described in this work produces in minutes what undergraduates in our course might initially take hours to days to perform, as they learn to adapt their existing computer science skills to the electronics and microcontroller domain. As such, in the near-term, systems such as WORDS2WIRES might be viewed as productivity assistants that allow (for example) scientists with existing coding skills but minimal electronics knowledge to quickly design instrumentation (such as data loggers) or other customized devices that are relatively modest in scope and complexity with a minimal time investment.

## 7    Conclusion

This study empirically characterizes the previously unknown potential of contemporary language models to move from *words* to *wires* – that is, to generate working electronic device designs from high-level text descriptions. Our analysis demonstrates these models have moderate-to-high proficiency in generating component-level knowledge on the PINS100 benchmark, while GPT-4 significantly outperforms CLAUDE-V1 at generating 25 fully-functional devices from the MICRO25 benchmark, reaching near-perfect performance. When used as a design assistant for generating six more complex devices, language models can generate devices that *nearly* meet specifications, but still require moderate correction by domain experts to function. While this novel application of language models inspires the democratization of electronic device creation, further development is currently tempered by the lack of simulators to automatically evaluate designs, and the highly manual nature of this process.

## 8    Limitations

This work has a number of limitations, including:

**Device scope:** The devices generated in this work are small in scope, with limited functionality – typically a small number of components, fewer than 50 lines of code, and controlled by ARDUINO microcontrollers which are frequently limited to only 2K of memory. This work does not address designing moderate or complex devices such as phones, personal computers, or other devices that are orders of magnitude more complex in terms of component counts and code length. For context, being able to successfully design all the devices in the MICRO25 benchmark would be equivalent to the performance of a particularly strong undergraduate student after having taken a first course in microcontroller design at our institution.

**Generation accuracy:** While the simple devices in Experiment 2 can reach high generation accuracy, particularly with GPT-4, nearly all devices in the more complex open generation condition in Experiment 3 contained errors, and required correction by a domain expert. In the open generation, three of six devices (*emoji keyboard*, *non-contact temperature sensor*, *ultrasonic glasses*) were generated in essentially functional forms in their base conditions (i.e. before adding additional requirements, such as

playing music when keys are pressed). A detailed error analysis is provided in APPENDIX D.

**Physical design and manufacture:** The physical manufacturing of the devices – including building circuits on prototyping breadboards using jumper wires, designing printed circuit boards, or designing physical 3D printed or laser cut enclosures, was entirely manual and completed by a human. While technologies (such as autorouting) exist to automate some of these aspects, they were not used in this work. Similarly, while language models have been shown to have some capacity for generating 3D object models (e.g. Nichol et al., 2022), that capacity has not yet developed to where it would be possible to generate enclosures or other mounting hardware required for physical device construction.

**Safety:** Constructing electronic devices has real dangers and potential harms, including but not limited to the risk of fire, electrical shock, and equipment damage, and should not be attempted by non-experts. The development environment is notoriously hostile to components, and even experienced electrical engineers frequently face safety challenges or accidentally destroy components. Generated devices should always be vetted by a domain expert, and not used for safety-critical applications, or applications where harmful unintended effects may be possible.

**Scope of Component Knowledge:** Popular electronics distributors in the US currently stock millions of different electronic components. Though many of those components belong to particular component classes that largely share pinout information (for example, DIGIKEY, a popular US-based distributor, lists approximately 1.5 million specific *resistors*, each with two pins), many of these – such as the 194,000 sensors currently available – do not generally share common pinouts or functions. The 100 common electronic components used in the PINS100 benchmark are representative of common components found on electronics tutorial websites, and that are frequently required to build basic digital microcontroller-controlled circuits – but are by no means an exhaustive set of the possible components available to construct electronic circuits.

**Speeding electronic design:** Just as coding assistants such as GITHUB COPILOT can increase human productivity for coding tasks (Ziegler et al., 2022), the use of a suite of electronic design assistants may similarly increase productivity in electronic device design, reducing the design process from days to hours (or, minutes). Currently, language models make a variety of errors on complex devices, and these errors are not always easy to predict. As such, the utility of language models as design assistants may be tempered in the near-term by the time required to manually review every aspect of a design for accuracy. As simulators and other automated evaluation methods become available, some of this burden of manual design review will decrease.

## Acknowledgements

We thank Peter Clark, Ashish Sabharwal, and the 4 anonymous reviewers for helpful comments on this work, as well as the Allen Institute of Artificial Intelligence (AI2) for funding this work.

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

## A    Experiment Hyperparameters

Models (GPT3.5, GPT-4, CLAUDE-V1) used similar hyperparameters through all experiments, with precise configurations and APIs available on the GITHUB repository. Models used greedy decoding *(temperature = 0)* to make experiments near-deterministic, though due to hardware-level implementation and other issues abstracted away though vendor APIs, model output still can change between successive runs. As such, specific (cached) model output for each experiment is provided in the GITHUB repository.

## B    Additional Experiment Details

### B.1    PINS100 Benchmark Evaluation

Two additional considerations complicate evaluation, which are addressed here. First, the same electrical component may come in different *packages*, or be built by different manufacturers – for example, the power pin (e.g. VCC) on a given component might be on pin 3 in one package, and pin 5 on a different package. As such, evaluation requires only that the pin name be correct (e.g. "VCC"), but does not require producing pin numbers, which are typically matched when choosing a particular component package during the circuit board design phase. Second, the specific names for pins are often described differently – and frequently with only single letters. For example, a pin with a *reset* functionality might be described variously as "RESET", "RST", or even simply "R" in different sources of text, such as official part datasheets or web tutorials, with each label being correct. As such, model-generated output is evaluated fully manually by a domain expert, who requires either generated pin names to match official documentation, obvious short forms, or (in the case of large differences) alternative part naming conventions found through a web search. This alignment process mirrors the actual electronics design process, where a reference schematic for a circuit may use different pin names than an official datasheet, and these pins need to be manually aligned by a domain expert by searching through a variety of reference materials.

#### B.1.1    Example of a non-critical pin

The permissive scoring metric for the PINS100 benchmark allows for missing or incorrect information for non-critical pins. A non-critical pin in this context is defined as a pin that is not, strictly speaking, required for basic use of a component. An example of a non-critical pin is the DRDY *(data-ready)* pin on an HMC5883L magnetometer. The magnetometer is an I2C device that nominally requires only 4 wires to function (if wired on a breadboard): SDA, SCL, VDD, and GND. The DRDY pin is marked as optional (not required) on the datasheet, as connecting the DRDY pin primarily allows faster polling rates by signifying when measurements are ready to be read by the host microcontroller.

### B.2    MICRO25 Benchmark Evaluation

Automatic evaluation of full devices faces a number of challenges, including that many different solutions are possible for each task, and existing simulators typically lack many of the possible components a model might generate in a solution. As such, evaluation was conducted manually by a domain expert. The schematics and code were manually inspected for functionality. When non-trivial or uncommon solutions were generated, the circuits were evaluated by constructing them in a simulator (AUTODESK TINKERCAD, shown in Figure 3) when possible, or physically building the circuits when not possible. When circuits used difficult to source or obsolete components, evaluation occurred through manual inspection, and comparing the generated schematics and code to reference materials. For a schematic to be considered correct, it must contain all relevant components, and be wired correctly in a way where code could be written to accomplish the desired task. For code to be considered correct, it must correctly perform the task given the generated schematic – or for cases where the schematic was incorrect, be able to accomplish the task as-is were the schematic corrected.

### B.3    Description of Human Evaluator

The domain expert used for evaluating this work is an author of this work, with the following qualifications: The evaluator is an award-winning science

educator, and prolific internationally-recognized open source hardware author with approximately 50 articles describing their open source hardware work in popular international news media such as Reuters, Forbes, and the Washington Post. The benchmarks described in this work were both authored and evaluated by the domain expert as reflective of the content of a popular full-term (4 month) undergraduate course in rapid prototyping and microcontroller design intended for computer science and information students, typically undertaken in a student's final year of undergraduate studies at an R1 ("very high research activity") university in the United States. The domain expert has delivered this course approximately 10 times to approximately 500 undergraduate students.

### B.4 Representation of Full Devices

*Additional information on device representation in Figure 3:* The *bill of materials* format expresses canonical information typical in the design process, including the component type (e.g. "resistor"), component name in the schematic (e.g. "R1"), component value (e.g. "10k ohms"), as well as a note on the purpose of the component (e.g. "current limiting resistor for LED"). The *pinouts* are expressed as a dictionary containing lists of pins for each part, as in Experiment 1. *Code* is expressed between MARKDOWN code blocks to ease extraction. *Schematics* are expressed as "netlists", which are a common storage format frequently adapted by electronic design tools. This format is analogous to an undirected graph, where edges represent a given connection from one component pin (such as the anode of an LED) to another component pin (such as one terminal of a current-limiting resistor).

### B.5 Description of 6 Open-Generation Devices

Six devices, crafted to use uncommon components, or common components in uncommon ways. The six devices are:

1. **Random number generator:** a random number generator, using two uncommon components: (a) a radiation sensor to help provide a random seed based on ambient radiation levels, and (b) a high-voltage nixie tube (or cold-cathode display), similar to a vacuum tube, to display randomly generated digits. Nixie tubes were manufactured and used in the 1950s and 1960s before light-emitting diodes became common.

2. **Emoji keyboard:** a USB keyboard that only contains keys for common emoji characters.

3. **Spectrometer:** a visible light spectrometer using the uncommon Hamamatsu microspectrometer, and that displays the spectrum on an organic LED (OLED) display.

4. **Non-contact temperature:** a device that measures the temperature of an object using a common infrared-based non-contact temperature sensor, but displays the temperature in an uncommon way: as a color-changing bar graph on an LED display.

In addition, two assistive devices were explored:

5. **Pill alarm:** a common pill-alarm, that displays the current time on an LCD display. The alarm is presented in an uncommon way: by using servo motors to physically waive flags that say *"take pill X"* for hearing-impaired users, until they press a reset button.

6. **Ultrasonic glasses:** common components (an ultrasonic distance sensor and piezo buzzer) used for an uncommon purpose – to create a pair of glasses for the visually-impaired. The glasses audibly notify the user of the distance to objects in front of them using a tone whose frequency varies with distance.

## C  Prompts

The full prompt for Experiment 1 (component pinouts) is provided in Table 3, while the full prompt for Experiments 2 and 3 (device generation) is provided in Table 6. All prompts are static – that is, the same format/n-shot examples shown here are also shown in every generation request – with the exception of the task strings ({**bolded**} in the tables), which are substituted with task-specific strings (representing the specific user-requested device to generate) at runtime. For Experiment 1, this is limited to the component name to generate pinout information for (e.g. *"DC motor"*). For experiments 2 and 3, this is limited to the target microcontroller platform (e.g. *"Arduino Uno"*) and plain-text device description (e.g. *"create a USB keyboard that only has buttons for the 9 most popular emojis on it"*).

```
Your task is to generate a description and pinout for an electronic component.
The specific electronic component to generate this output for is: {componentName}
The output format is JSON, between code blocks, as shown in the example below:
'''
{
    "7479": {
        "description": "Dual D positive-edge triggered flip flop, asynchronous preset and clear",
        "pinout:"["#R1", "D1", "CLK1", "#PR1", "Q1", "#Q1", "VSS", "#Q2", "Q2", "#PR2", "CLK2", "D2", "#R2", "VDD"]
    }
}
'''
```

Table 3: The prompt for Experiment 1 (component pinout generation) on the PINS100 benchmark. The specific 1-shot example in the format prompt (the the 7479 FLIP FLOP) is used across every experimental run – only the string representing the target device to generate pinout information for ({componentName}, e.g. "DC MOTOR") changes.

## D   Error Analysis: Modifications to Open Generation Devices

### D.1   High-level Qualitative Challenges

Overall high-level qualitative challenges in designing the six case study devices are described briefly below, where a detailed description of errors and corrections for each of the six devices is provided in APPENDIX D.2.

**Sensitivity to prompt:** Small and seemingly helpful changes in the device description or prompt can cause large changes in generation. For example, for the *random number generator*, including a reminder that the radiation sensor required a pull-up resistor appeared to cause the model to forget to include a high-voltage supply for the nixie tube.

**Many requirements can create poor performance:** Adding many composite requirements to a project, even when they are individually easy, can create low performance. For example, for the *emoji keyboard*, adding the requirement to play a relevant musical tune when each key is pressed generally produced only scaffolds for music generation code without actually including the melodies. A subsequent call to GPT-4 asking it only to fill in this music scaffold was required to generate the melodies.

**Device drivers:** The model performs best for straight-forward circuits where the coding portion of interfacing with external components (such as sensors) is abstracted to existing libraries. When writing a low-level device library is required, the model commonly either hallucinates a non-existent library, or generates a reasonable first-pass at a device library that requires extensive modification to function. For example, the *microspectrometer* device driver the model generated had the essential conceptual-level components – i.e. that data needed to be clocked out of the spectrometer and read by an analog-to-digital converter after sending a start pulse to the spectrometer – but the generated code had incorrect clock timings, logic levels, and other fine-grained details which made it unable to function without correction.

**Deprecated or mismatched libraries:** Hardware libraries are frequently updated to support new features, but the ARDUINO ecosystem lacks *Makefiles* or a build manager with explicit library version numbering, making specific library versions unknown for a given code example. During manufacture, this frequently required searching through old versions of a library (such as the *Bounce* button input library for the *emoji keyboard*) to find a version that matched the specific API generated by the model. Similarly, observing examples of different APIs across different library versions causes the model to occasionally mix APIs from older and newer versions of libraries, or from different libraries with similar functions (such as combining the APIs of several LCD display libraries for the *spectrometer*).

**End-of-life parts:** Electronic components regularly reach end-of-life cycles, and are no longer manufactured or easily available. The model occasionally generated circuits that used unavailable parts, and had much less competency generating circuits for newer part variants, particularly those that were released near GPT-4's knowledge cutoff date of September 2021 (OpenAI, 2023).

### D.2   Device-Specific Errors

The devices generated in the open generation experiment generally required modification to function as intended. Here we provide a list of the major design changes required to reach functionality:

### D.2.1 Random number generator

Both the radiation sensor (*Radiation Watch Type 5*) and the nixie tubes (*IN-12A*) are highly uncommon components, and are likely to have limited examples available in existing documentation. Generally, across several iterations of device description prompts, either the radiation sensing circuit was correct, or the nixie tube circuit was correct, but not both. The radiation detector requires a pull-up resistor to function, and is pulled low when a high-energy particle strikes it. The nixie tube requires an external high-voltage driver, which was usually generated correctly, but when generated incorrectly it was typically powered by USB voltage (5V) instead of the required high-voltage (170V). Across device descriptions, the method used in the code to set the random seed based on the radiation sensor varied – some useful, some largely incorrect.

### D.2.2 Emoji keyboard

The base emoji keyboard was largely generated without issue, though did mix up the version for the button input library, and failed to mention the special programming requirements for the *Teensy* microcontroller to place it in human-interface-device (HID) mode to act as a USB keyboard. The specific emojis were chosen by the model, adding only the requirement that they must be high-frequency, and at least one must be the heart emoji. Adding many requirements generally reduced design quality – for example, adding the requirement that some of the emojis needed to be at least 5 ASCII characters long was not generally successful (and, the model occasionally generated emojis that were unicode, which is generally easily supported by the USB HID standard, or occasionally generated only single characters instead of full emojis). Similarly, adding the requirement for a short musical tune to play upon pressing an emoji, where the tune should have a similar affect to the emoji (e.g. a love song for the heart emoji, a happy song for the happy emoji, etc.) generally produced only harsh single tones, or the scaffold for generating the music without actual musical tones for each emoji. This scaffold was provided to GPT-4 on its own in a post-generation step, and the resultant code added to the original code.

### D.3 Visible Spectrometer

This code had two central challenges: generating a device driver for an uncommon component (the Hamamatsu C12666MA micro-spectrometer), and using a library for a common component (a display with a common controller). Using different device descriptions, the model either hallucinated non-existent libraries for interfacing to the spectrometer, or generated its own libraries that had the high-level procedure correct (e.g. sending a start pulse to the spectrometer, then continuously sending a clock pulse while reading data using an analog-to-digital converter to read out each of the 256 spectral channels) – though the specifics of the device driver, such as timing or logic levels were typically incorrect and needed to be manually corrected. With respect to the display, three OLED and TFT displays with common display controllers were attempted, and the most successful (the 128x128 OLED using a SSD1351 controller) was used. There were only two small errors in the display code: the initial call to the display had reversed the order of the arguments, and the last call to the display (swapping the backbuffer) was for a different library, and not required here.

### D.4 Non-contact temperature sensor

This device consisted of two common components (an MLX90614 non-contact temperature sensor, and an 8-pixel neopixel RGB LED strip). The device generated without issue, and was only modified slightly to reverse the direction the LED bar graph displayed from (to accommodate the mounting constraints of the specific LED strip used).

### D.5 Pill alarm assistive device

This device consisted of three common components: an 16x2 LCD with an I2C interface, three hobby servos, and a single pushbutton. The schematics generated largely without issue. The code generally had a number of logic errors that needed correction when the added requirement of oscillating the flags back-and-forth was added, including that the code would oscillate all flags, regardless of which alarm (e.g. pill 1, pill 2, or pill 3) was active. Different generated instances of this device in response to different specifications either kept track of time internally, or used an external realtime clock module for more accurate timekeeping – but all generated devices failed to provide any means of setting the initial time of the device other than manually in code, which is an important usability feature of a clock not explicitly mentioned in the prompt.

| Category | Task Description String |
|---|---|
| Digital - Button | Create a device with a single push button, that turns on an I/O pin when the button is pressed, and turns off that same I/O pin when the button is not pressed. |
| Digital - Multiple | Create a device with 4 push buttons, that turns on an I/O pin when exactly 2 of the buttons button are pressed, and turns off that same I/O pin otherwise. |
| Analog | Create a device that reads the analog input from a potentiometer configured as a voltage divider and sourced by 5V. When the input is greater than or equal to 2.5V, it should turn on an I/O pin, while if the input is below 2.5V it should turn off the I/O pin. |
| **Protocols** | |
| Serial/UART | Create a device that reads the Serial port at 9600 baud. Whenever the string "hello" is transmitted to the device, it will respond by sending "world". |
| Serial/SPI | Create a device that connects to another device using SPI. Whenever the string "hello" is transmitted to the device, it will respond by sending "world". |
| I2C | Create a device that reads a value from an I2C device, and displays that value to the serial port every second. The device address is 0x50, the register to read is 0x15, and the value is 8 bits long. The value should be displayed in base 10. |
| SD Card | Create a device that opens a file called "out.txt" on an SD card, and and every 10 seconds, randomly prints one of the following animal names (as well as a newline character): cat, dog, mouse, parrot. |
| **Output** | |
| Motor - DC | Create a device that oscillates between spinning a DC motor one direction then the other direction every 5 seconds. |
| Motor - Stepper | Create a device that oscillates the output of a stepper motor clockwise then counterclockwise 45 degrees, every 5 seconds. |
| Motor - Servo | Create a device that contiuously moves a hobby servo back and forth from 45 degrees to 135 degrees every 5 seconds. |
| LED - Blink | Create a device that blinks an LED every 500 milliseconds. |
| LED - Sequence | Create a device with 4 LEDs, that blink in sequence, one after the other, every 500 milliseconds. When the end of the sequence is reached, the pattern should reset, and continue indefinitely. |
| LED - 7 Segment | Create a device that counts from 0 to 9 on a 7 segment display, indexing numbers every 500 milliseconds. When the end of the cycle is reached, it should start again. |
| LED - Neopixel | Create a device with a neopixel (WS2812) that slowly and continuously cycles a rainbow of colors. |
| Relay | Create a device that turns a relay on for 2 seconds, then off for 5 seconds, continuously. The relay coil takes 500 milliamps of current to engage. |
| LCD | Create a device that prints the phrase "Hello World" on a 16x2 LCD. Use a HD44780 compatible 16x2 LCD, configured normally (i.e. without an I2C, Serial, or other simpler connection). |
| Sound - Buzzer | Create a device that continuously plays a high, medium, then low tone on a Piezo Buzzer, changing tones every second. |
| Analog Output | Create a device that produces a sawtooth wave, ramping up from 0V to 5V every 50 milliseconds. |
| **Sensors** | |
| Resistive - CDS | Create a device that reads the value from a CDS cell, and outputs it to the Serial port every second. |
| Manual Protocol | Create a device that reads the distance from a HC-SR04 ultrasonic distance sensor, and outputs the distance (in centimeters) to the Serial port every second. |
| I2C - Magnetic | Create a device that reads the current magnetic field readings using a HMC5883L magnetometer. The readings (x, y, z, and total field strength) should be output to the Serial port every second. |
| **Logic** | |
| Simon | Create a device that implements the popular memory game simon, where users enter progressively longer sequences of colors. It should have 4 possible colors, and include sound when the button is touched, as well as when winning/losing. The game should timeout if the user doesn't enter input after 5 seconds. |
| Conway | Create a device that implements the popular Conway's Game of Life, on an 8x8 LED matrix. The game steps should cycle every 500 milliseconds. If the board is empty, it should randomly initialize the game again. There should be a pushbutton that allows randomly resetting the game. |
| Clock | Create a device that implements a clock that prints the current time on a 16x2 character LCD display with an I2C interface. It should have three buttons to help set the time in a user freidlly way: one to increment hours, one to increment minutes, and one to increment seconds. The timekeeping should be performed by the Arduino, and not an external real-time clock. |
| Air Temperature | Create a device that reads the current air temperature, and displays it as a color on a neopixel (WS2812). The color should be fully blue at 0C, fully red at 30C, and an interpolation of blue and red between those temperatures. |

Table 4: The full *(i.e. not truncated or summarized)* task strings used for the MICRO25 benchmark. The target platform for each is the ARDUINO UNO microcontroller.

## D.6 Ultrasonic glasses assistive device

This device generated with only small issues. The device requirements specified using a specific battery-powered ESP32 microcontroller board, but the schematic used digital pin numbers that were unavailable on this specific microcontroller board – these were assigned to other pins trivially. The library the model used for sound generation (TONE) is famously available for most Arduino devices except the ESP32, and was modified to use a different function specific to the ESP32 with a similar signature, but with two added initialization and termination calls. The specific audio frequency range generated by the model was also modified to a reduced range and more fitting for human ears, as the original included high-frequency tones that, while audible, were uncomfortable and resembled a fire alarm.

## E  Device Descriptions: 25 Benchmark Tasks and 6 Open-generation Devices

The full device task description strings for the MICRO25 benchmark are shown in Table E. A set of iterated task description strings for the open-generation condition (Experiment 3) are provided in Table 5. These and additional task descriptions are provided in the GITHUB repository.

| Device | Task Description String |
|---|---|
| Random number generator | Radioactive dice: a device that uses the radiation rate from a radiation watch type 5 sensor (which outputs a digital signal, active low, depending on whether a high-energy particle has struck it at that moment or not) to determine the random seed for an electronic dice. The device should continually read the radiation sensor, accumulate the count, and use it to help change the random seed periodically. Every 3 seconds, the device should display the roll of a 6 sided dice on a Nixie tube. It should use an IN-12 nixie tube, and K155ID1 driver. |
| Emoji USB keyboard | Create a keyboard that plugs in as a USB device, but instead of a full keyboard it has only a small number of buttons. The keyboard should only have buttons for 9 popular emojis, expressed as ASCII characters, not unicode. One emoji should be a heart. There should be an LED that's on all the time, but blinks off for 500 milliseconds when a button is pressed. There should also be a piezo buzzer, that plays a brief tune that is of the same affect as the emoji being pressed – for example, a love song for the heart emoji, a happy song for a happy emoji, sad music for a sad emoji, and so forth. |
| Visible light spectrometer | Create a visible spectrometer that continuously displays the spectrum on an OLED display. It should use the Hamamatsu C12666MA 5v-compatible mini-spectrometer for the spectrometer (pins: 5V, GND, EOS, START, CLK, GAIN, VIDEO). The display should be a 128x128 pixel OLED with a SSD1351 controller and SPI interface, also 5V compatible (pins: GND, VIN, CD, MISO, SDCS, OLEDCS, RESET, DC, SCK, MOSI). |
| Non-contact temperature sensor | Create a non-contact temperature sensor using the MLX90614. The temperature should be output on a 8-pixel neopixel strip. 0 degrees or below should light only the first neopixel. For each 10C after, another neopixel should light. The color of the neopixel should change according to its temperature (blue=cold, green=mild, yellow=warm, orange=warmer, red=hot). |
| Pill alarm | Create a pill alarm. The alarm should have a clock that prints the current time on a 16x2 character LCD display. If the time is 6:30am, noon, or 6:30pm, the device should raise one of 3 flags (signifying different pills need to be taken). Servo 1 controls flag 1, servo 1 controls flag 2, and servo 3 controls flag 3. When raised, the servo should move from 0 degrees to 90 degrees. The servo should stay up until a button is pressed, after which it's reset to the down position (0 degrees). While raised, the servos should slowly oscillate between 45 and 90 degrees, to help get the user's attention. |
| Ultrasonic Glasses | Glasses for the blind that provide a helpful sound that corresponds to how close something is in front of them. Should have a slide switch that can disable the sound. Please use the MaxSonar ultrasonic distance sensor. |

Table 5: Example iterated task description strings for the 6 open-generation devices in Experiment 3. A set of initial task descriptions was progressively iterated, expanded, and refined based on task performance, before arriving at the above task descriptions.

You are DeveloperGPT, the most advanced AI developer tool on the planet. You answer any coding question, and provide real useful example code using code blocks. Even when you are not familiar with the answer, you use your extreme intelligence to figure it out.
Further, you have specialized training in electronics, and can design embedded electronic circuits based around the {microcontrollerPlatformStr} platform, coupled with programs to make those circuits successfully accomplish tasks.
Your task is to:
{taskStr}

Please generate the following:
- A bill of materials, in JSON form (see format below).
- A pinout, in JSON form (see format below). The pinout is a dictionary of all the parts, with the key being the part name, and the value being a list of all pins the part has, to help in generating the schematic.
- A schematic, in JSON form (see format below). Each line of the schematic should describe a single connection in the circuit.
- A complete {microcontrollerPlatformStr} program that implements the program to successfully complete the task.
Each section should be between code blocks '''.
- A brief set of special instructions, in point form, if required.

Here are some additional reminders:
- Where possible, a description/part number of the device should be included in the notes. Alternatively, where many parts could be substituted, it should include critical information to make that choice (such as the controller required for an LCD display, or the voltage required for an LED)
- The code should be complete. It can #include built-in {microcontrollerPlatformStr} libraries, but otherwise should contain all the code to compile and run as-is.

Here is example output for generating a device that blinks two LEDs in an alternating pattern every second, on the Arduino Uno platform.

Bill of materials:
'''
```
[
    {"part":"Arduino Uno", "name":"uno", "value":"", "notes":"Arduino Uno microcontroller"},
    {"part":"LED", "name":"D1", "value":"red", "notes":"alternating LED 1. Standard voltage range (2-3.3V)."},
    {"part":"LED", "name","D2", "value":"white", "notes":"alternating LED 2. Standard voltage range (2-3.3V)."},
    {"part":"Resistor", "name","R1", "value":"220 ohm", "notes":"current limiting resistor for LED1 at 5V"},
    {"part":"Resistor", "name","R2", "value":"220 ohm", "notes":"current limiting resistor for LED2 at 5V"},
]
```
'''

Pinouts: '''
```
{
    "Arduino Uno": ["5V", "3.3V", "GND", "AREF", "D0/RX", "D1/TX", "D2", "D3", "D4", "D5", "D6", "D7", "D8", "D9", "D10",
        "D11", "D12", "D13", "A0", "A1", "A2", "A3", "A4/SDA", "A5/SCL"],
    "D1": ["anode", "cathode"],
    "D2": ["anode", "cathode"],
    "R1": ["1", "2"],
    "R2": ["1", "2"]
}
```
'''

Schematic (list of connections):
'''
```
[
    [{"name":"D1", "pin":"cathode"}, {"name": "uno", "pin":"GND"}], # Connect D1 cathode to Uno GND
    [{"name":"D1", "pin":"anode"}, {"name": "R1", "pin":"2"}], # Connect D1 anode to pin 2 of R1 (current limiting resistor)
    [{"name":"R1", "pin":"1"}, {"name": "uno", "pin":"D5"}], # Connect pin 1 of R1 (current limiting resistor) to
                                                             # Uno Digital I/O 5 (D5), to activate/deactivate D1
    [{"name":"D2", "pin":"cathode"}, {"name": "uno", "pin":"GND"}], # Connect D2 cathode to Uno GND
    [{"name":"D2", "pin":"anode"}, {"name": "R2", "pin":"2"}], # Connect D2 anode to pin 2 of R2 (current limiting resistor)
    [{"name":"R2", "pin":"1"}, {"name": "uno", "pin":"D6"}], # Connect pin 1 of R2 (current limiting resistor) to
                                                             # Uno Digital I/O 5 (D6), to activate/deactivate D2
]
```
'''

Arduino Uno Code:
'''
```
// Alternating blink
// This code interfaces with a circuit that has two LEDS that blink in an alternating pattern.
// The pattern changes every second.

// LED 1 on Digital I/O 5
#define PIN_LED1 5
// LED 2 on Digital I/O 6
#define PIN_LED2 6

// the setup function runs once when you press reset or power the board
void setup() {
    // Initialize LED pins to output mode
    pinMode(PIN_LED1, OUTPUT);
    pinMode(PIN_LED2, OUTPUT);
}

// the loop function runs over and over again forever
```
(Prompt continues onto next page...)

```
(Prompt continued from previous page page...)
void loop() {
    digitalWrite(PIN_LED1, HIGH); // Turn LED 1 ON
    digitalWrite(PIN_LED2, LOW); // Turn LED 2 OFF
    delay(1000); // wait for a second
    digitalWrite(PIN_LED1, HIGH); // Turn LED 1 OFF
    digitalWrite(PIN_LED2, LOW); // Turn LED 2 ON
    delay(1000); // wait for a second
}
'''
Instructions:
'''
- This code uses only standard libraries. No additional libraries are required in the library manager.
- Assemble circuit and program as normal.
'''
Snippet examples (also for the Arduino Uno):
—
Example: Connecting a servo
Bill of Materials:
'''
[
    {"part":"Servo Motor", "name":"S1", "value":"", "notes":"Standard 3-wire 5V compatible hobby servo (e.g. SG90)"}
]
'''
Pinouts:
'''
{
    # Arduino Uno omitted for space in snippet
    "Servo Motor": ["VCC", "GND", "signal"]
}
'''
Schematic (list of connections):
'''
[
    [{"name":"S1", "pin":"signal"}, {"name": "uno", "pin":"D3"}], # Connect Servo 1 signal to Uno D3
    [{"name":"S1", "pin":"VCC"}, {"name": "uno", "pin":"5V"}], # Connect Servo 1 VCC to Uno 5V
    [{"name":"S1", "pin":"GND"}, {"name": "uno", "pin":"GND"}] # Connect Servo 1 GND to Uno GND
]
'''
—
Example: Connecting a button (pull-up)
Bill of Materials:
'''
[
    {"part":"Button", "name":"BT1", "value":"", "notes":"Momentary push button"},
    {"part":"Resistor", "name":"R1", "value":"10k ohm", "notes":"Pull-up resistor for button"}
]
'''
Pinouts:
'''
{
    # Arduino Uno omitted for space in snippet
    "Button": ["1", "2"],
    "Resistor": ["1", "2"]
}
'''
Schematic (list of connections):
'''
[
    [{"name":"BT1", "pin":"1"}, {"name": "uno", "pin":"D2"}], # Connect Button pin 1 to Uno D2
    [{"name":"BT1", "pin":"1"}, {"name": "R1", "pin":"1"}], # Connect Button pin 1 to R1 pin 1
    [{"name":"R1", "pin":"2"}, {"name": "uno", "pin":"5V"}], # Connect R1 pin 2 to Uno 5V (pull-up)
    [{"name":"BT1", "pin":"2"}, {"name": "uno", "pin":"GND"}] # Connect Button pin 2 to GND
]
'''
—
Example: This is a case of what NOT to do.
Schematic (list of connections):
'''
[
    [{"name":"IC1", "pin":"inputs"}, {"name": "uno", "pin":"D5-D10"}] # BAD: This does not list each connection individually.
                                              # It is not clear which pin on the IC is connected to which pin on the Uno.
] '''
—
Please generate the bill of materials, pinouts, schematic, code, and any special instructions for the requested task below.
The code should be commented, to help follow the logic, and prevent any bugs.
The platform is: {microcontrollerPlatformStr} .
The task is: {taskStr} .
```

Table 6: The full prompt for device generation, which includes one full positive device example (*a simple device that blinks two LEDs in an alternating pattern*), as well as two positive snippets (*illustrating a component with more than 2 pins (servo), and a connection with more than 2 components (pull-up resistor), respectively*), and one negative snippet designed to promote explicitly enumerating schematic connections. At runtime, **{microcontrollerPlatformStr}** is replaced with the platform (e.g. Arduino Uno), and **{taskStr}** is replaced with the description of the target device to generate (e.g. from Tables 4 or 5).