# OpenReview forum: "From Words to Wires: Generating Functioning Electronic Devices from Natural Language Descriptions"
_EMNLP/2023/Conference — EMNLP 2023 Findings_

### Official Review · Reviewer_zUFP · 2023-07-31

**Soundness:** 3

**Excitement:**

3: Ambivalent: It has merits (e.g., it reports state-of-the-art results, the idea is nice), but there are key weaknesses (e.g., it describes incremental work), and it can significantly benefit from another round of revision. However, I won't object to accepting it if my co-reviewers champion it.

**Paper Topic And Main Contributions:**

This paper empirically evaluates LLMs' ability to generate electronic designs based on high-level and freeform language descriptions. Three experiments are described that test the model's capability in 3 tasks with increasing difficulty.

The main contributions are
- The evaluation task in a domain that is not accessible to the typical NLP researcher. The task targets a series of steps are needed to (later) generate a final product.
- The application and evaluation of the task using the GPT-4 and Claude models.

**Questions For The Authors:**

- Question A: In experiment 2, are the 5 types of tasks tested only in isolation? Are any of them related in a way that they would be carried out sequentially by humans? Or would humans carry them out in any order (I am referring to both types of tasks and the specific tasks listed in the table)?

- Question B: What is the relation between the device descriptions in Figure 4 and Appendix B4? Are either of these the input/query descriptions?

- Question C: Were there any issues with the models producing the correct formal output syntax, e.g. valid json or json according to the (prompt) instructions? Are these included in the failure cases when calculating accuracy? It would be interesting to separate this from actual wrong or faulty specifications.

**Reasons To Accept:**

- The work evaluates LLMs in a domain that many NLP experts are unfamiliar with but that is an obvious use case for LLMs that is similar to code generation. Particular domain knowledge is needed to carry out this evaluation, making this paper a great interdisciplinary contribution.
- The paper is well structured and written and the three experiment setups are well chosen to investigate the question.

**Reasons To Reject:**

- The evaluation is only superficial with no further insights into interesting failure (or success) cases. Experiment 1 and 2 are evaluated via accuracy. The evaluation of experiment 3 does not contain any quantification of the problematic cases that are mentioned. Appendix D contains more details but also no numbers.

- It is difficult for me to understand the difference of this work to code generation, mainly because I cannot understand the relation of the different outputs to one another (see also Question section). Figure 2 shows 4 different types of output and this diagram suggests they would be generated in order but it is not clear whether they depend on one another and if yes, how the evaluation shows this.

**Reproducibility:**

2: Would be hard pressed to reproduce the results. The contribution depends on data that are simply not available outside the author's institution or consortium; not enough details are provided.

**Reviewer Confidence:**

4: Quite sure. I tried to check the important points carefully. It's unlikely, though conceivable, that I missed something that should affect my ratings.

**Typos Grammar Style And Presentation Improvements:**

- About presentation, I find the many images of devices a bit misleading because no physical objects are being generated. While the images are illustrative, they add little without further explanation. It is impossible for me, who has no background knowledge in electronics, to understand how the formal specifications relate to the images. I only see the relation between free-form input language and image, e.g. for the emoji keyboard, but would need more explanation for the relation between specs and image. In Figure 1, I have no idea at all what I am looking at (I googled nixie tube and then was able to recognize it). Since the focus is on generating specifications, as a reader I would rather like to understand the specs better, even if textual and formal specs seem more boring than images (some of the images can stay of course, or be smaller, or move to the appendix). For example, it is more meaningful to me to see Figure 2 with the 4 boxes illustrating the generation task than Figure 1 that hints at generating an actual circuit. I would recommend merging the two figures.

- Instead of images (or instead of large images since they are taking quite a bit of space) I think one of the concrete examples from the appendix would be great in the main body of the paper e.g. a truncated version of Table 6.

- When referring to appendix sections, please refer to the specific section. If you are using the LaTeX template, this can be done by using labels just like section labels. It helps to find information faster.

- p. 4 "Results": "is shown Table 1" --> "is shown in Table 1."

- GPT-4 is sometimes spelled GPT4 throughout the paper

- Table 2: mentions "popular" games that in fact might not be known so much by everyone, maybe you could provide links to descriptions

- Appendix D, title: "Modifcations" -> "Modifications"

- D.1, intro is mentioning the appendix but this is the appendix.

- In the main body of the paper, I felt that the important domain-specific vocabulary was explained or exemplified where necessary. In the appendix at times, I could not follow the argumentation because I lack domain knowledge. For example in B1 probably because this wasn't detailed in the paper. I just have no idea of the concept of a pin and what it entails, even with this explanation.

- I liked that all the figures and tables are well formatted with easy-to-read captions and other text!

---

> ### Author Rebuttal · Authors · 2023-08-28
>
> We thank the reviewer for their detailed and thoughtful review.
>
> **Rebuttal to Reasons to Reject:**
>
> 1. **Evaluation follows best-practices, not superficial:**\
> **Reviewer 4:** "The evaluation is only superficial with no further insights into interesting failure (or success) cases. Experiment 1 and 2 are evaluated via accuracy. The evaluation of experiment 3 does not contain any quantification of the problematic cases that are mentioned. Appendix D contains more details but also no numbers."\
> **Response:** The evaluation (using Pass@1, not accuracy) is the standard evaluation based on contemporary research methods for code generation, and the main metric used for most benchmarks including HumanEval from Codex (Chen et al., 2021).  Further, examining detailed coding errors is not standard practice -- even the massive 54 page StarCoder paper ( https://arxiv.org/pdf/2305.06161.pdf ), unconstrained by page limits, includes only a single short paragraph (Page 29, Section 8.3, **154 words**) on the utility of using their model as an assistant, noting that it is "somewhat capable yet brittle", and that "[it] has clear limitations: it sometimes proposes wrong solutions, presents wrong facts, and can make offensive comments".  **Even still, in contrast to other contemporary work, this work includes a substantially larger error analysis than even StarCoder at 2.5 pages (~ 1500 words) in the Appendix**.  While it is true that the error analysis doesn't include a table of quantitative categories, this is not standard practice for case studies because the total number of samples in each bin would be too low to be helpful (i.e. statistically significant using inferrential statistics).  Instead, case studies -- especially for papers proposing new tasks that require expansive domain-expert evaluation (like medical tasks) -- typically provide a detailed exposition of case studies to motivate their position and analysis, and promote future research in evaluation methods.
>
> 2. **Clarity on differences with respect to code generation tasks:**\
> **Reviewer 4:** "It is difficult for me to understand the difference of this work to code generation, mainly because I cannot understand the relation of the different outputs to one another (see also Question section). Figure 2 shows 4 different types of output and this diagram suggests they would be generated in order but it is not clear whether they depend on one another and if yes, how the evaluation shows this."\
> **Response:** The main difference between code generation and this device-generation task is that this task requires both code generation and the new task of electrical schematic generation (and, these must be done jointly, so that the code can use the schematic -- e.g. the code trying to turn on an LED on Arduino D3 while the LED is on D4 would not be helpful).  The bill-of-materials and component-pinouts are provided to mirror how humans design circuits (i.e. first you need to know the parts you'll use (bill of materials), and what pins they have (component pinouts) before you can connect them electrically (schematic) and write code that accomplishes the task (code)).  **We welcome (and strongly encourage!) thoughts with respect to clarifying the presentation of the work for a joint NLP + Electrical Engineering audience, and have indeed workshopped a preprint of this paper through a moderate audience to improve its clarity to a broad audience, but do not believe that fixable concerns with clarity on an interdisciplinary paper warrant a hard-reject (2 on soundness).** Similarly, though our task and use case are brand new (and, we believe, truly transformative for de-skilling electronic device design to people with no background in electrical engineering), the ACL reviewing guidelines ( https://2023.aclweb.org/blog/review-acl23/ ) explicitly state that new and interdisciplinary research should not be discounted because of its novelty -- indeed this is what we should be branching towards: "Heuristic: This has no precedent in existing literature.  Why this is problematic: Believe it or not: papers that are more novel tend to be harder to publish. Reviewers may be unnecessarily conservative."
>
>
> **Clarification Response to questions for authors:**
>
> **Question A: In experiment 2, are the 5 types of tasks tested only in isolation? Are any of them related in a way that they would be carried out sequentially by humans? Or would humans carry them out in any order (I am referring to both types of tasks and the specific tasks listed in the table)?**\
> **Response:** The prompt for Experiment 2 (nominally, what's shown in Figure 3) produces 4 kinds of output: the (a) bill of materials (BOM), (b) component pinouts, (c) electrical schematic, and (d) code.  Of these, (a), (b), and (c) are directly related because (1) every part on the BOM has to be on the schematic, and (2) every component pinout has to be on the schematic.  In this way, we include BOM and pinout generation as a way of promoting "chain of thought" prompting (Wei et al., 2022), explicitly asking the model to perform the task in the same pipeline that humans perform it so that much of the information it needs to generate the schematic is already present in the generation.  We describe this in Section 4 ("Representations" subsection, L275-L289), but would welcome thoughts on improving this text for a broader audience. \
> Similarly, one need only evaluate the schematic to fully evaluate the BOM and the component pinouts (which is why we evaluate only the schematic and code directly).  But, this would not be intuitive to those without an electronics background, and hasn't been caught by  our previous proofreading -- thank you for pointing this out, we will clarify it in the text.
>
> **Question B: What is the relation between the device descriptions in Figure 4 and Appendix B4? Are either of these the input/query descriptions?**\
> **Response:** **Part 1:** The 6 device descriptions in Figure 4 are the same as those 6 described in more detail in the Appendix B4 (i.e. "Random Number Generator" in Figure 4 is the same as B4, "Emoji USB keyboard" in Figure 4 is the same as "Emoji Keyboard" in B4, etc).  **Part 2:** While the device descriptions have much of the same content as the actual task prompts to generate the devices in the model, they are edited for length.  All task descriptions used to generate all devices in the paper are included in the Github repository.  For example, the task string for the USB keyboard task was:
>
> ```
> [keyboard-emoji6a] create a keyboard that plugs in as a USB device, but instead of a full keyboard it has only a small number of buttons. The keyboard should only have buttons for 9 popular emojis, expressed as ASCII characters, not unicode.  One emoji should be a heart.  There should be an LED that's on all the time, but blinks off for 500 milliseconds when a button is pressed.  There should also be a piezo buzzer, that plays a brief tune that is of the same affect as the emoji being pressed -- for example, a love song for the heart emoji, a happy song for a happy emoji, sad music for a sad emoji, and so forth.
> ```
>
> **Question C: Were there any issues with the models producing the correct formal output syntax, e.g. valid json or json according to the (prompt) instructions? Are these included in the failure cases when calculating accuracy? It would be interesting to separate this from actual wrong or faulty specifications.**\
> **Response:** This is a good question.  There were zero errors as a result of generating output in an incorrect format.  This is both because the models are very good to adhering to JSON format prompts, and the few formatting issues they had in pilot experiments (like adding comments that said `"# ... And so on"` in generating schematics) were entirely resolved by including a few example generations in the prompt.  While this does make the prompt large (~ 2k tokens, see Table 6 in the Appendix), the benefit is that it's free of generating any syntax errors. We will add this clarification re: zero syntax errors in the text.
>
>
>
> **Typos Grammar Style And Presentation Improvements:**
>
> Thank you for sending these comments on presentation.  One of the challenges with writing this paper is that the work is extremely exciting, but it's also for a new and highly interdisciplinary task that blends a field that one normally doesn't see combined with NLP (electrical engineering).  We have already workshopped the paper through a large number of pure-NLP colleagues, and will use your comments to further refine our descriptions.
>
>
> **Remarks on reproducibility:**
>
> This is a work of open source science, and every effort has been made to make the resulting research products easy to use and well-documented for reproducibility.  The code, benchmarks, libraries, and results currently sit in a Github repository awaiting being made public.  Runscripts allow users to regenerate the results in about 12 hours (at current API speeds).  Unlike other tasks that require evaluation by domain experts (e.g. medical tasks, where doctors might disagree that a given symptom is individative of one diagnosis versus another), in our tasks the evaluations are objective -- i.e. the LED either blinks, or it doesn't; the servo motor either oscillates back and forth between 45 and 135 degrees, or it doesn't.  Papers that require (a) domain expertise, or (b) API access are still considered highly replicable (the PIs advisors used to have to travel across the world 20 years ago to access specific Linguistic Data Consortium (LDC) corpora to replicate others work).  Where possible and applicable, the results of this work are augmented with additional information (e.g. screenshots of the simulator, that augment the JSON netlists produced by the model) for usability and to quickly aid in verification.
>
> The reproducibility score suggests this work is not reproducible, but we believe this to be in error as every effort has been made to make this work  reflective of *highly reproducible* open source science.

---

### Official Review · Reviewer_8oFb · 2023-08-10

**Soundness:** 1

**Excitement:**

2: Mediocre: This paper makes marginal contributions (vs non-contemporaneous work), so I would rather not see it in the conference.

**Missing References:**

- Figure 3 should be closer to page 5 where it is referred to.


**Paper Topic And Main Contributions:**

This paper introduces two benchmarks, related to text-to-code tasks: one for assessing a language model's capability to generate electronic component specifications from a brief textual requirement, and one for assessing the more complex task if fully designing a micro-controller circuit including a list of materials, the component's pinouts, the schematic, and code, from a brief textual requirement. All non-code output is represented in the json format. The datasets consist of 100 and 25 test instances each.

The paper examines the performance of pretrained language models (GPT-3.5, GPT-4, and CLAUDE-V1) on the proposed benchmarks, and shows they achieve up to 86% and 94% pass@1 scores respectively. The paper also includes six case studies over more challenging inputs, but those are not directly evaluated, but instead corrected by an expert and constructed as physical devices.

**Questions For The Authors:**

A. Could you provide an example over which exact parts are considered critical and non-critical on strict vs. permissive evaluation.

B. Is ground truth or references also provided in the benchmarks?

C. The high performance of the models seems to indicate that the pretrained language models have been exposed to such problems during training. This is also indicated by the authors in lines 337-345. Did the authors investigate on what extend the pretrained language models could have been exposed to inputs and outputs such as those included in the benchmarks?


**Reasons To Accept:**

- The paper presents benchmarks on a novel task, related to text-to-code synthesis.

**Reasons To Reject:**

- The high performance of the models suggest that the benchmarks contain little challenge for future work to investigate and expand upon. This could indicate that the task is trivial or that the dataset is not representative of the task's difficulty. The case study inputs are claimed to be more challenging, but those are not properly evaluated.
- It is unclear how the evaluation was conducted, and whether it is possible to be reproduced consistently by future work.
- Lines 317-321, and elsewhere, suggest that this was a form of human evaluation by an expert. In that case more information should be included in the paper, e.g. was it a single expert, what is their expertise and their relation to the paper, what steps were taken to eliminate their bias in the study, and if and how was inter-annotator agreement determined.
- The adoption of the pass@k metric, which was proposed for evaluating the functionality of code over a set of unit tests, seems inappropriate for these benchmarks. In essence, what is measured seems more akin to accuracy or exact match scores. Given the nature of the output, n-gram overlap metrics may be appropriate (provided that the benchmark also provides references).
- The merits of the case study are very unclear, specifically how physically manufacturing expert-corrected devices can quantify the pretrained language models' capabilities.


**Reproducibility:**

1: Could not reproduce the results here no matter how hard they tried.

**Reviewer Confidence:**

4: Quite sure. I tried to check the important points carefully. It's unlikely, though conceivable, that I missed something that should affect my ratings.

---

> ### Author Rebuttal · Authors · 2023-08-29
>
> We respectfully strongly disagree with this review:
>
> - **Code (and electrical schematics) cannot be evaluated with n-gram overlap:**\
> Code **can not** be evaluated using n-gram overlap with reference code, because there are many possible solutions that produce the same output -- this is why code is evaluated by running it, and testing its output.  The exact same situation applies to electrical circuits -- there are many possible schematics that can produce a given output.  There are no existing methods to perform unit tests on hardware generation tasks such as this, so evaluation must be completed manually by a domain expert.
>
> - **GPT-4 performing well at a benchmark does not prohibit publication**:\
> This does not follow precident.  GPT-4 meets or exceeds human performance on many popular benchmarks -- for example, GPT-4 answers (and explains) nearly all 4th and 8th grade standardized science exam questions correctly, yet we still use these standardized benchmarks to measure model performance -- and so too should we on electronics tasks, where strong performance would reflect being a highly successful student (95%+ percentile) in our undergraduate microcontroller class.  Similarly, while it's accepted that GPT-4 outperforms nearly all other models on nearly every task, as we see in this paper, even the next-most-performant model (Claude-V1) achieves only 60% on schematics and 76% on code, while open models are surely much less capable.  Indeed, GPT4 performance is often seen as a ceiling to be pursued for open source models (such as the plethora of code generation models released only in the past 2 weeks seeking to meet GPT4 performance on HumanEval), not a sign to discontinue research in a subfield.
>
> - **Evaluation details are provided throughout; experiments completely reproducible**:\
> "It is unclear how the evaluation was conducted, and whether it is possible to be reproduced consistently by future work".  Details of the experiments and evaluation are provided throughout the paper and Appendix.  As described in the paper (Introduction, Appendix), all data, code, results, and runscripts are provided in a Github repository that will be made public upon acceptance, and the data can be regenerated in about 12 hours (plus whatever time the domain expert requires for their evaluation).  While this domain requires expert evaluation, that evaluation is highly objective to a domain expert: a circuit either successfully blinks and LED, or it doesn't; a servo motor either successfully oscillates between 135 and 45 degrees, or it doesn't.  The results are eminently reproducible, and if in doubt, you are invited to submit the prompt from Table 6 to GPT-4 8k, substituting in the following platforms and  task descriptions from MICRO25 (TSV below), to voice any valid criticisms of the objectivity of the evaluation empirically:
>
> ```
> micro25.tsv
> generalArea	specificArea	platform	taskName	taskDescription
> Input	Digital - Button	Arduino Uno	input-digital-button	Create a device with a single push button, that turns on an I/O pin when the button is pressed, and turns off that same I/O pin when the button is not pressed.
> Input	Digital - Multiple Buttons	Arduino Uno	input-digital-multiple-buttons	Create a device with 4 push buttons, that turns on an I/O pin when exactly 2 of the buttons button are pressed, and turns off that same I/O pin otherwise.
> Input	Analog - Potentiometer	Arduino Uno	input-analog-potentiometer	Create a device that reads the analog input from a potentiometer configured as a voltage divider and sourced by 5V.  When the input is greater than or equal to 2.5V, it should turn on an I/O pin, while if the input is below 2.5V it should turn off the I/O pin.
> Protocols	Serial	Arduino Uno	protocols-serial	Create a device that reads the Serial port at 9600 baud.  Whenever the string "hello" is transmitted to the device, it will respond by sending "world".
> Protocols	I2C	Arduino Uno	protocols-i2c	Create a device that reads a value from an I2C device, and displays that value to the serial port every second.  The device address is 0x50, the register to read is 0x15, and the value is 8 bits long.  The value should be displayed in base 10.
> Protocols	SPI	Arduino Uno	protocols-spi	Create a device that connects to another device using SPI.  Whenever the string "hello" is transmitted to the device, it will respond by sending "world".
> Protocols	SD Card	Arduino Uno	protocols-sd	Create a device that opens a file called "out.txt" on an SD card, and and every 10 seconds, randomly prints one of the following animal names (as well as a newline character): cat, dog, mouse, parrot.
> Output	Motor - DC	Arduino Uno	output-motor-dc	Create a device that oscillates between spinning a DC motor one direction then the other direction every 5 seconds.
> Output	Motor - Stepper	Arduino Uno	output-motor-stepper	Create a device that oscillates the output of a stepper motor clockwise then counterclockwise 45 degrees, every 5 seconds.
> Output	Motor - Servo	Arduino Uno	output-motor-servo	Create a device that contiuously moves a hobby servo back and forth from 45 degrees to 135 degrees every 5 seconds.
> Output	LED - Blink	Arduino Uno	output-led-blink	Create a device that blinks an LED every 500 milliseconds.
> Output	LED - Sequence	Arduino Uno	output-led-sequence	Create a device with 4 LEDs, that blink in sequence, one after the other, every 500 milliseconds.  When the end of the sequence is reached, the pattern should reset, and continue indefinitely.
> Output	LED - Neopixel	Arduino Uno	output-led-neopixel	Create a device with a neopixel (WS2812) that slowly and continuously cycles a rainbow of colors.
> Output	Relay	Arduino Uno	output-relay	Create a device that turns a relay on for 2 seconds, then off for 5 seconds, continuously.  The relay coil takes 500 milliamps of current to engage.
> Output	LCD - Raw	Arduino Uno	output-lcd-raw	Create a device that prints the phrase "Hello World" on a 16x2 LCD.  Use a HD44780 compatible 16x2 LCD, configured normally (i.e. without an I2C, Serial, or other simpler connection).
> Output	Sound - Buzzer	Arduino Uno	output-sound-buzzer	Create a device that continuously plays a high, medium, then low tone on a Piezo Buzzer, changing tones every second.
> Output	7 Segment Display	Arduino Uno	output-7segment	Create a device that counts from 0 to 9 on a 7 segment display, indexing numbers every 500 milliseconds.  When the end of the cycle is reached, it should start again.
> Output	Analog Output	Arduino Uno	output-analog	Create a device that produces a sawtooth wave, ramping up from 0V to 5V every 50 milliseconds.
> Sensors	Resistive - CDS Light Resistor	Arduino Uno	sensors-resistive-cds-cell	Create a device that reads the value from a CDS cell, and outputs it to the Serial port every second.
> Sensors	Manual Protocol - Ultrasonic	Arduino Uno	sensors-ultrasonic	Create a device that reads the distance from a HC-SR04 ultrasonic distance sensor, and outputs the distance (in centimeters) to the Serial port every second.
> Sensors	I2C Protocol - Magnetic	Arduino Uno	sensors-magnetic	Create a device that reads the current magnetic field readings using a HMC5883L magnetometer.  The readings (x, y, z, and total field strength) should be output to the Serial port every second.
> Logic	Interactive - Simon Game	Arduino Uno	logic-simon	Create a device that implements the popular memory game simon, where users enter progressively longer sequences of colors.  It should have 4 possible colors, and include sound when the button is touched, as well as when winning/losing.  The game should timeout if the user doesn't enter input after 5 seconds.
> Logic	Non-interactive - Conway's Game of Life	Arduino Uno	logic-conway	Create a device that implements the popular Conway's Game of Life, on an 8x8 LED matrix.  The game steps should cycle every 500 milliseconds.  If the board is empty, it should randomly initialize the game again.  There should be a pushbutton that allows randomly resetting the game.
> Logic	Interactive - Clock	Arduino Uno	logic-clock	Create a device that implements a clock that prints the current time on a 16x2 character LCD display with an I2C interface.  It should have three buttons to help set the time in a user frieldly way: one to increment hours, one to increment minutes, and one to increment seconds.  The timekeeping should be performed by the Arduino, and not an external real-time clock.
> Logic	Interactive - Air Temperature	Arduino Uno	logic-air-temperature-meter	Create a device that reads the current air temperature, and displays it as a color on a neopixel (WS2812). The color should be fully blue at 0C, fully red at 30C, and an interpolation of blue and red between those temperatures.
> ```
>
> While the paper mentions (in no less than 13 separate locations, by our count) details of the domain expert evaluation, we will include the additional requested details of the domain expert in the Appendix: "The electrical engineering expert used for evaluating this work is a co-author of this paper, award-winning science educator, prolific open source hardware author, and internationally-recognized researcher with approximately 50 articles describing their open source hardware work in popular international news media such as Reuters, Forbes, and the Washington Post.  The benchmarks described in this work were both authored and evaluated by the domain expert as reflective of the content of a popular full-term (4 month) undergraduate course in microcontroller design for computer science students, typically taken in a student's final year of undergraduate studies at a R1 ("Very high research activity") institution in the United States.  The domain expert has delivered the course approximately 10 times over a decade, to approximately 500 students.  Students in this computer science course have competed in a design competition against engineering students -- and won each year -- validating the scope and quality of this curriculum."
>
> **Questions For The Authors:**
>
> - **Question A: Example of non-critical pin:**\
> *Response:* An example of a non-critical pin would be the DRDY (data-ready) pin on an HMC5883L magnetometer.  The magnetometer is an I2C device that nominally requires only 4 wires to function (if wired on a breadboard): SDA, SCL, VDD, and GND.  The DRDY pin is marked as optional (not required) on the datasheet, as connecting the DRDY pin only allows faster polling rates by signifying when measurements are ready to be read by the host microcontroller.  We will include such an example in the Appendix.
>
> - **Question B: Are ground-truth/references provided in the benchmarks:**\
> *Response:* For code generation/schematic generation tasks, there are no ground truths -- many solutions are possible, which is why their output (rather than their code) is evaluated.  For example, even the most trivial task (blinking an LED) could place the LED on any of 20 I/O pins, use any resistor in the range of 100 ohms to 470 ohms, use a variety of different colors of LED with different forward voltages, write the code in a way that directly through libraries (e.g. digitalWrite()) or indirectly through registers (e.g. PORTB) controls the I/O pin, that delays through blocking (e.g. delay() ) or non-blocking (e.g. watchdog timer/interrupts) methods, etc. -- and this covers only canonical solutions.  A non-canonical solution might control the LED through a transistor, opto-coupler, or relay to allow a wider voltage range, or even a specialized LED driver.  Indeed, we have observed that GPT-4 tends to prefer canonical solutions, while Claude-V1 prefers to create unusual solutions (presumably from first principles).  Given the open-ended, non-expert nature of the user interface (i.e. we want the user to be able to say "Make me a device that blinks an LED every 5 seconds", not "Make me a device that connects an LED with a 2V forward voltage to pin D6 for PWM control, through a 220 ohm current limiting resistor, with code that uses the digitalWrite() interface"), making highly-constrained references is nether possible nor desirable.
>
> - **Question C: The high performance of GPT-4 suggests similar tasks may be present in the training data.  Did the authors examine this?**
> *Response:* Yes, we exactly examined this, as one of the primary motiviations for Experiment 3.  Quoting the first sentence of Section 5 ("Experiment 3: Open Device Generation"): **"While we’ve observed that language models have the capacity to design comparatively simple devices in Experiment 2, this result is tempered by these benchmark tasks being representative of fairly common skills and capacities that are frequently taught in microcontroller-oriented curricula found in books, internet tutorials, and blog posts – and as such, the MICRO25 tasks likely exist in some form in the voluminous (but closed) training data of these models. Here, we examine how well the best-performing model, GPT-4, can create comparatively more complex devices in a more realistic and qualitative setting [...]. To further increase task difficulty, each of the device specifications in this experiment were explicitly crafted to be highly unusual – either using uncommon components, or combining common components in unusual ways – such that the likelihood of similar devices appearing in the closed model training data is low. The six devices are shown in Figure 4."**
>
> While the training data to GPT-4 is closed, we can infer that the relative difference in performance between Experiment 2 and Experiment 3 (i.e. 24/25 benchmark tasks generated without error, versus 2/6 case studies generated without error) suggests that training data may play a large part of the performance difference.  Though, as detailed in the **2.5 page detailed error analysis for the case studies provided in the Appendix**, performance is affected by many factors, from intuitive factors (such as that adding many subtasks, such as keys playing music for the Emoji keyboard, affects performance) to unintutive factors (e.g. prompt sensitivity -- i.e. small changes in the prompt causing GPT-4 to fail to include a high-voltage power supply for the Nixie Tube display in the random number generator).  We can similarly infer that GPT-4 has never seen the datasheet or driver for the Hamamatsu C12666MA microspectrometer (Appendix D.3), given that all the timings it generated in the device driver were incorrect, but it clearly has seen linear image sensors before, because they all follow similar protocols for clocking data out, and the driver was mostly correct at a conceptual level if not at a timing level.
>
> **Overall comments:**
>
> Our overall concerns for this review -- providing a score of *1* for what, objectively, is -- as Reviewer 2 said best -- a paper that "pushes the boundaries of what NLP can achieve, showing the potential in the field of NLP extends far beyond routine NLP tasks" and that "... has the potential to democratize electronic device design, making it available to those without a deep expertise but with a clear idea of vision", is succinctly described by a colleague not involved with this work but who read this review:
>
> ```
> It is maddening to me that a well-evaluated incremental piece of work, where datasets and methodologies are plentiful but the work is incredibly boring, is rated more highly than imaginative and innovative work blazing a new trail – where by definition evaluation needs to be rethought. It’s a common problem with reviewing, to the extent that ACL has explicitly instructed reviewers not to succumb to this bias ( https://2023.aclweb.org/blog/review-acl23/ ).
> ```
>
> Is generating the designs (either directly, or through an assistant interface) for full electronic devices purely from high-level natural language descriptions of those devices an exciting, enabling technology?  We think so.
>
> Does this work perform sound, well-evaluated, reproducible science?  We would robustly defend "yes".  Does this paper perform work at scale that is representative of the field?  As the contemporaneous references provided by Reviewers 1 and 2 show (Thakur et al., 2023; Blocklove et al., 2023), the scale of this work and evaluation is *three times* larger than contemporaneous work.  We would remind potential reviewers that this is the /first/ paper for this subfield, not the last paper, and by definition it takes time for research methods and infrastructure to be rethought to enable the work at great scale.

---

### Official Review · Reviewer_bhLT · 2023-08-10

**Typos Grammar Style And Presentation Improvements:** I think it would be a lot better to a…
**Soundness:** 2

**Excitement:**

3: Ambivalent: It has merits (e.g., it reports state-of-the-art results, the idea is nice), but there are key weaknesses (e.g., it describes incremental work), and it can significantly benefit from another round of revision. However, I won't object to accepting it if my co-reviewers champion it.

**Justification For Ethical Concerns:**

None.

**Missing References:**

None.

**Paper Topic And Main Contributions:**

This paper delves into the groundbreaking exploration of LLM capabilities, particularly in the field of electronic devices design. Traditionally, LLMs have been praised for their proficiency in understanding and generating textual content. However, this research reveals a new aspect of these models: the ability to design functional electronic circuits based on high-level textual descriptions. This capability has also been confirmed by some recent work, such as Chip-Chat (https://arxiv.org/abs/2305.13243).

Main Contributions:

1. Benchmarks Introduction: The authors introduced two distinct benchmarks to assess the abilities of language models in electronic design: PINS 100 This benchmark evaluates the model's understanding of various electrical components. MICRO 25: A more comprehensive benchmark that assesses a model's capability to design common microcontroller circuits, accompanied by relevant code within the Arduino ecosystem.

2. Performance Analysis: The paper presents a thorough performance analysis, noting that models such as GPT-4 and Claude-V1 achieved impressive results, ranging from 60% to 96% on the task of generating complete devices.

3. Case Studies: Six detailed case studies are provided, showcasing the potential of language models as design assistants for moderately intricate devices. These case studies not only serve as practical demonstrations of the model's capabilities but also offer insights into the real-world applications of such a technology.

4. Qualitative Analysis: The authors delve deep into a qualitative assessment of model performance, highlighting the challenges in evaluation and offering recommendations for future advancements in complex circuit design.

Essentially, this paper stands at the crossroads of natural language processing and electronic design, presenting a new research perspective. Thanks to the authors for showing us the untapped potential of some LLMs.


**Questions For The Authors:**

First and foremost, I commend the authors for their innovative approach to utilizing LLMs. The paper is well structured and the contributions are interesting. However, in order to fully understand the scope, implications and future direction of this research, I have several questions:

Question A: Benchmarks: Can you elaborate on the metrics used to design benchmarks PINS 100 and MICRO 25? How are these benchmarks validated to ensure they adequately measure the capabilities they are designed to test?

Question B: Error analysis: The paper mentions error analysis for open generation experiments in an appendix. Can you highlight the most common types of errors encountered and speculate on their underlying causes?

Question C: Human intervention design: Given that the physical creation of equipment is manual, how do you foresee the integration of automated manufacturing processes in this pipeline? Is it feasible to have language models to guide such processes in the future?

Looking forward to your answers, they will help me understand the research more fully.

**Reasons To Accept:**

Innovative Applications: This article shows a novel application of NLP, which is more innovative in transitioning from pure text processing to tangible electronic device design.

Empirical Results: Empirical results, especially the GPT-4 model, performed well on benchmarks such as PINS 100 and MICRO 25.

Introduction of new benchmarks: The introduction of benchmarks such as PINS 100 and MICRO 25, which provide standardized evaluation metrics for evaluating models in this novel application, will undoubtedly serve as the basis for future research.

Beneficial case studies: The inclusion of six case studies not only validates the thesis's claims, but also provides practical insights into the practical application of the proposed methodology.

Comprehensive analysis: This paper not only highlights the advantages and potential of language models in electronic design, but also acknowledges their limitations, opening a clear path for future research.

Benefits to the NLP community:

Expanding horizons: This paper pushes the boundaries of what NLP can achieve, showing that the potential in the field of NLP extends far beyond routine NLP tasks.

Implications for interdisciplinary research: By combining NLP with electronic design, this work can inspire further interdisciplinary research.

Resource Contribution: The introduced benchmarks will be an invaluable resource, enabling researchers to evaluate and improve the state-of-the-art in this emerging field.

Democratizing Design: If further refined, the paper's approach has the potential to democratize electronic device design, making it available to those without deep electronics expertise but with a clear idea or vision.

In conclusion, this paper is a good way to encourage people to think outside the traditional scope of NLP and consider how to use NLP to empower various fields.

**Reasons To Reject:**

While this article describes the new application of LLMs in electronic devices, there are several issues that make me question:

1. Design of textual descriptions: The generation of high-level descriptions is done by experts, and the LLMs essentially knowledge is code generated, which makes me question the potential of end-to-end automation.

2. Simple device range: The complexity of the devices selected for this article is limited, most devices have less than 50 lines of code and are controlled by a microcontroller with 2K memory. This gives me some concerns about practical applicability and scalability.

3. Human intervention: Almost all devices contain errors that require the intervention and correction of domain experts. However, the uncertainty of human intervention to reproduce the entire job can become very large.

4. Lack of comparative analysis: While this paper presents benchmarks and presents results, there is a lack of comparison with other potential approaches and no in-depth exploration of why the use of LLMs is superior or necessary for this task.

In conclusion, while this paper touches on the innovative intersection of NLP and electronics, more comprehensive research, broader scope, and rigorous validation are needed.

**Reproducibility:**

3: Could reproduce the results with some difficulty. The settings of parameters are underspecified or subjectively determined; the training/evaluation data are not widely available.

**Reviewer Confidence:**

3: Pretty sure, but there's a chance I missed something. Although I have a good feel for this area in general, I did not carefully check the paper's details, e.g., the math, experimental design, or novelty.

---

> ### Author Rebuttal · Authors · 2023-08-28
>
> We thank the reviewer for their detailed and thoughtful review.
>
> **Summary of rebuttal:**
>
> 1. **Review text is a 5, review score is a 2 (Review score does not match text).** \
> To quote this review:
> - "pushes the boundaries of what NLP can achieve, showing the potential in the field of NLP extends far beyond routine NLP tasks"
> - "the introduced benchmarks will be an invaluable resource, enabling researchers to evaluate and improve the state-of-the-art in this emerging field"
> - "this papers approach has the potential to democratize electronic device design, making it available to those without a deep expertise but with a clear idea of vision"
>
> **We showed this review to 4 other colleagues not involved with this work, who all said the score based on the text should be a 5 (strong accept), not a 2 (hard reject).  We are frankly stunned at the score.**
>
> *Additional quotes:*
>
> "thorough performance analysis", "impressive results", "detailed case studies", "practical demonstrations of the model's capabilties [and] also offer insights into the real-world applications of such a technology", "[also delve deep into a qualitative assessment of model performance", "novel application of NLP", "will undoubtedly serve as the basis for future research", "six case studies not only validate the thesis' claims, but also provide practical insights into the practical application of the proposed methodology", "opens a clear path for future research"
>
>
> **Rebuttal to Reasons to Reject:**
>
> 1. **"Design of textual descriptions: The generation of high-level descriptions is done by experts, and the LLMs essentially knowledge is code generated, which makes me question the potential of end-to-end automation."** \
> *Response:* This is not true -- the high-level descriptions of the devices are in natural language, not expert generated (e.g. for the emoji keyboard, it's `create a keyboard that plugs in as a USB device, but instead of a full keyboard it has only a small number of buttons. The keyboard should only have buttons for 9 popular emojis.`).  We will clarify this in the description.
>
> 2. **Simple device range: The complexity of the devices selected for this article is limited, most devices have less than 50 lines of code and are controlled by a microcontroller with 2K memory. This gives me some concerns about practical applicability and scalability.** \
> *Response:* This is a criticism of the entire field of microcontroller-based electronics, not of this paper.  **Nearly every single non-trivial electronic device created after the 1990s contains one or more microcontrollers, and nearly all of those microcontrollers contain less than 2k of RAM.**  Embedded applications are bare-metal (i.e. without an operating system) and do not generally have large RAM requirements -- indeed, higher RAM only became available in microcontrollers when they started to be used for digital signal processing (e.g. audio) applications.  A keyboard, a mouse, indeed a visible light spectrometer all have modest RAM requirements and easily fit.  **Indeed, the ATMega328p targeted in this work is widely believed to be the most popular microcontroller in the world.** \
> \
> Similarly, "most devices [requiring generating] less than 50 lines of code" is not a valid criticism -- the entire field of code generation could not fit 50 lines of code into the entire context length of its best models only 6 months ago!  HumanEval (Chen et al., 2021), the most popular code-generation benchmark, generally requires generating less than 5 lines of Python code per problem.  *Generating 50 lines of error-free microcontroller code in C in response to a short natural language task description is a truly incredible achievement, like from science fiction!*
>
> 3. **Human intervention: Almost all devices contain errors that require the intervention and correction of domain experts. However, the uncertainty of human intervention to reproduce the entire job can become very large.**\
> *Response:* This is also a criticism of the entire field of code generation, not this paper.  **Every known system (e.g. the extremely popular Github Copilot coding assistant) is deployed as an assistant that requires a human to check and correct the work.**  Further, our system shows extremely high performance without assistance (Experiments 1 and 2), and uses a domain expert to quantify performance on very challenging devices (Experiment 3), much as a Github Copilot would be used in real-world scenarios.
>
> 4. **Lack of comparative analysis: While this paper presents benchmarks and presents results, there is a lack of comparison with other potential approaches and no in-depth exploration of why the use of LLMs is superior or necessary for this task.**\
> *Response:* **Re: comparison**, to the best of our knowledge, this is the very first paper do perform this task, so it is not possible to compare against previous work.  **Re: motivation**, we provide the following text in the paper (L056-L061): *"Our focus in this work is on automating the task of transforming a high-level concept into a practical electronics schematic, complete with companion microcontroller code—a task that currently requires significant human expertise and effort.".*  We can expand this to highlight the massive time savings (e.g. our undergraduates typically take several days to design and complete a Simon Says memory game (from MICRO25) in an Arduino -- the system reported in this paper can do this in under a minute).
>
>
> **Clarification Response to questions for authors:**
>
> **Question A: Benchmarks: Can you elaborate on the metrics used to design benchmarks PINS 100 and MICRO 25? How are these benchmarks validated to ensure they adequately measure the capabilities they are designed to test?**\
> *Response:* This is a good question.  The PINS100 benchmark components are validated by being all high-frequency (i.e. commonly occurring) core components found from electronics simulators and popular tutorial websites (e.g L197-L198). The MICRO25 benchmark are validated by all being core skills taught across many undergraduate microcontroller curricula (including a popular undergraduate course taught for a decade by the authors).
>
> **Question B: Error analysis: The paper mentions error analysis for open generation experiments in an appendix. Can you highlight the most common types of errors encountered and speculate on their underlying causes?**\
> *Response:* The error analysis in the Appendix already includes these, but we could highlight these in the main body of the paper using the extra page, if that's what you're asking?  For example, GPT4 favors using common parts and libraries, while Claude-V1 likes to design things manually from first principles (which is much more difficult), which may contribute to the difference in performance.
>
> **Question C: Human intervention design: Given that the physical creation of equipment is manual, how do you foresee the integration of automated manufacturing processes in this pipeline? Is it feasible to have language models to guide such processes in the future?**\
> *Response:* This is a good question -- and we do indeed foresee this (which is why the excitement of this work is so high).  It is easily foreseeable that using a model that constraints its generation to the components available in a large pick-and-place machine, that is combined with a circuit board mill, printer, or other device, that one could go from natural language description to physically-manufactured-device in less than an hour within a few years time.
>
> **Missing references:**
>
> Thank you for pointing us to Chip-Chat (Blocklove et al., Arxiv 22 May 2023), which is considered a contemporaneous work by ACL policy ( https://www.aclweb.org/adminwiki/index.php/ACL_Policies_for_Submission,_Review_and_Citation ). We will include it in our related work.  That being said, comparing our work to the work you have suggested:
>
> 1. *25 vs 8 tasks, task complexity:* We introduce two benchmarks (one with 100 components, another with 25 designs), whereas Chip-Chat's benchmark includes only 8 designs (pg. 3 Table 1), many of them far simpler than our benchmarks (e.g. an 8-bit shift register; though he dice roller is nearly comparable to our random number generator without the radiation sensor for the random seed, or the nixie tube driver).
>
> 2. *Verilog vs Arbitrary Hardware:* Chip-Chat is working in Verilog for FPGAs, and being Verilog, is essentially a highly-constrained code-generation task evaluable entirely with unit tests, versus our open-ended hardware generation task that necessitates physical construction and expert evaluation.
>
> 3. *6 vs 1 case study:* Chip-Chat paper includes a single case study where the system is used in the role of an assistant (the 8-bit accumulator-based processor), our paper includes *6*.
>
> 4. *Paper length requirements*: The IEEE (?) format allows the Chip-Chat paper to have what we measured to be 11k words (50k characters). The ACL format gives us ~ 5500 words (~30k characters), approximately half that of the Chip-Chat paper.  While our paper introduces benchmarks that are between 3 to 6 times greater in scope than ChipChat, the ACL paper length requirements necessitate that some details (like the error analysis) must be in the Appendix.
>
> **Remarks on Reproducibility:**
>
> This is a work of open source science, and every effort has been made to make the resulting research products easy to use and well-documented for reproducibility.  The code, benchmarks, libraries, and results currently sit in a Github repository awaiting being made public.  Runscripts allow users to regenerate the results in about 12 hours (at current API speeds).  Unlike other tasks that require evaluation by domain experts (e.g. medical tasks, where doctors might disagree that a given symptom is individative of one diagnosis versus another), in our tasks the evaluations are objective -- i.e. the LED either blinks, or it doesn't; the servo motor either oscillates back and forth between 45 and 135 degrees, or it doesn't.  Papers that require (a) domain expertise, or (b) API access are still considered highly replicable (the PIs advisors used to have to travel across the world 20 years ago to access specific Linguistic Data Consortium (LDC) corpora to replicate others work).  Where possible and applicable, the results of this work are augmented with additional information (e.g. screenshots of the simulator, that augment the JSON netlists produced by the model) for usability and to quickly aid in verification.
>
> The reproducibility score suggests this work is not reproducible, but we believe this to be in error as every effort has been made to make this work  reflective of *highly reproducible* open source science.
>
>
> **Updated Scores:**
> Based on the truly extraordinary praise in the original review, along with the detailed comments addressed above, we request the reviewer update their scores from hard reject (2/3) to accept.

---

### Official Review · Reviewer_P9mH · 2023-08-11

**Soundness:** 3

**Excitement:**

3: Ambivalent: It has merits (e.g., it reports state-of-the-art results, the idea is nice), but there are key weaknesses (e.g., it describes incremental work), and it can significantly benefit from another round of revision. However, I won't object to accepting it if my co-reviewers champion it.

**Missing References:**

The paper could be improved by providing additional discussion and references related to using LLMs for circuit design, such as:

* Thakur, Shailja, et al. "Benchmarking Large Language Models for Automated Verilog RTL Code Generation." 2023 Design, Automation & Test in Europe Conference & Exhibition (DATE). IEEE, 2023.

**Paper Topic And Main Contributions:**

The paper studies the capabilities of commercial LLMs to provide components for the design of electronic devices starting from textual descriptions. To that end, the paper also releases two relevant datasets (PINS100 and Micro25) that include relevant data required microelectronics design that span all required components to design fully functional microelectronic devices. The paper also proposes a set of evaluation metrics for evaluating the performance of LLMs in microelectronic design and showcases six real-world use cases of devices built from LLM outputs. The authors first describe the general problem setting and their contributions in the introduction and subsequently provides a discussion of related work in arduino devices and LLM based code generation.

Next, the authors describe their first set of experiments and analysis where they assess the capability of LLMs to generate accurate pinout information for 100 common electronic components included in their PINS100 dataset. The second experiment focuses on circuit generation, which constitutes a significantly more complex task that involves four different element that jointly form the full circuit. The authors assess different LLMs' capabilities with a PASS@1 metric and provide an analysis of the various LLMs across the different design cases included in Micro25 dataset.

The authors final set of experiments involve creating real-world devices based on the output of different LLMs following several rounds of refinement of specification prompts, which are then created by a human using common manufacturing processes for circuits. The authors show pictures of the six different devices and describe their general capabilities, as well as a series of challenges related to correcting different errors in the process through a human expert. Lastly, the authors provide a discussion of different challenges and opportunities (prompt sensitivity, manual evaluation, infusing LLMs with common sense knowledge) along with a conclusion and an extensive discussion of limitations.

**Questions For The Authors:**

* Question A: You say that the LLMs are instruction fine-tuned. What kind of instructions did you use and how?
* Question B: Are there any procedures for what happens when the LLM suggests a part that is not available?
* Question C: You said that the devices were synthesized by a domain expert that had to correct some common errors. You also state that the LLM's capabilities are comparable to a student from an undergraduate course in microelectronics. Does the LLM reach undergraduate capabilities with or without the corrections? I am still to get a better understanding of the true capabilities of the LLMs.
* Question D: Can you provide the prompts that you used to design the six real world use cases?

**Reasons To Accept:**

* The paper provides a framework for design real-world electronic devices based on natural language instruction and goes through the entire process ending up with six different physical devices.
* The paper generally defines evaluation metrics and experiments well defined and laid out in clear and understandable manner.
* The dataset contributions could be useful for the community to build upon.
* The authors ensure that limitations, cataloging of different of errors, and the importance of human expertise is well-represented for different aspects of evaluation and construction of physical devices.

**Reasons To Reject:**

* While the paper includes a discussion on code generation with LLMs, it does not touch on code design relevant to electronic circuits specifically. It would be good to get a better understanding of how the paper distinguishes itself from the relevant literature in that aspect.
* The authors could have provided a more thorough analysis of the time saving provided by using LLMs. This is especially true in cases where iterative refinement of inputs was needed.
* The utility of the Micro25 dataset is limited if human evaluation is still required to assess performance.
* Some additional details on how the LLMs were used would be helpful.
* Since human expertise is required in various parts of the process, reproducibility is more challenging.

**Reproducibility:**

2: Would be hard pressed to reproduce the results. The contribution depends on data that are simply not available outside the author's institution or consortium; not enough details are provided.

**Reviewer Confidence:**

3: Pretty sure, but there's a chance I missed something. Although I have a good feel for this area in general, I did not carefully check the paper's details, e.g., the math, experimental design, or novelty.

---

> ### Author Rebuttal · Authors · 2023-08-28
>
> We thank the reviewer for their detailed and thoughtful review.
>
> Summary of rebuttal in the context of reasons to reject:
>
> 1. **Code design relevant to electronic circuits:** While it is true that our Related Work does not include a detailed discussion of code generation applied to electronic circuits specifically, that is because (to the best of our knowledge) this paper is one of the first works in this new subfield.  ACL policy ( https://www.aclweb.org/adminwiki/index.php/ACL_Policies_for_Submission,_Review_and_Citation ) defines contemporaneous works as those within a 3 month period before the submission window (i.e. after March 2023). We will of course update our related work to include all work published in the interim between submission and camera ready deadlines, but *ACL policy forbids this being used as a reason for rejection: "In such cases, reviewers should point authors to the non-cited work (so that they can discuss it in the camera-ready version) but not penalize the authors for missing the citation."*
>
> 2. **Expert evaluation as a negative:** *Expert evaluation performed by a human domain expert is the *gold-standard* research methods and evaluation for NLP, not a reason for rejection.*  Proxy metrics (e.g. BLEU/ROUGE in Machine Translation) are used as inexpensive but inaccurate approximates of performance when expert evaluation is too expensive, and developed only when the field has progressed enough to develop and validate such metrics (e.g. even though character-level overlap was used in the original IBM alginment papers (e.g. pg 84 of Brown et al., 1990, "A statistical approach to language translation", Computational Linguistics), it wasn't until 14 years later (Lin, WAS 2004, "Rouge: A package for automatic evaluation of summaries") that these metrics became studied in detail and popularized).  Other subfields still haven't made that progression -- *for example, no one would reject a Medical NLP paper because the task required expert evaluation by doctors -- similarly, rejecting a paper in a new subfield because it also requires domain expertise (electrical engineering) for evaluation is unfounded.*  ACL policy ( https://2023.aclweb.org/blog/review-acl23/ ) cites that "Having no precedent in existing literature: Believe it or not: papers that are more novel tend to be harder to publish. Reviewers may be unnecessarily conservative." is not grounds for rejection.  \
> **Additionally:** The experiments performed in this work were done to the highest standards of natural language processing research methods.  The code, benchmarks, library, and experimental results are committed to a github repository, awaiting public release.  \
> **Objectivity:** While code generation (and, circuit generation) can be notoriously difficult to automatically evaluate because: (a) there are many possible valid code and circuit solutions for a given task, (b) there are no current ways to write unit tests for arbitrary microcontroller hardware (e.g. no simulator supports arbitrary components an LLM might mention, or even more mundane but useful parameters such as reporting what angle a simulated servo is at), the *tasks are objective and straightforward for a domain expert to evaluate*.  Where two doctors might disagree that an elevated liver enzyme is indicative of two different diagnoses, for this task, the measures are completely objective -- i.e. the LED either blinks, or it doesn't; the servo either oscillates back and forth between 45 and 135 degress, or it doesn't; the correct distance is either reported on the serial console for the HC-SR04 ultrasonic distance sensor, or it isn't.  This is essentially the first paper in this subfield, not the last, and future work by our team and others will work to make evaluation faster with more capable simulators (which are a huge amount of infrastructure to stand up; but we're working on it), and in the mean time examining whether new proxy metrics are a viable alternative.
>
> 3. **Analysis of time savings vs human effort:** The PI on this project says: "While we definitely agree it would be nice to have some large human user study on the time savings associated with using large language models for device design, measuring this is not straightforward, and indeed not even yet possible for coding assistants.  For example, even the Github Copilot team (arguably the most-deployed and most-used coding assistant) have difficulty quantifying measurements of productivity just for code assistants ( https://github.blog/2022-09-07-research-quantifying-github-copilots-impact-on-developer-productivity-and-happiness/ ), and are currently using survey-based measures (e.g. "Do you feel more productive when using Github Copilot?", "Do you feel you can focus on more satisfying work when using Github Copilot?", "Do you feel you use less mental effort on repetitive tasks when using Github Copilot?").  It would not be possible to solve the research methods problems of quantifying code assitant contributions to productivity (and, running a large user study) in the one additional page afforded by EMNLP, but we look forward to exploring the human-centered computing aspects of this in future work, and submitting those papers to appropriate (e.g. HCI) venues.  What we can likely say from our experience with teaching undergraduates these same concepts is a similar story to Copilot: that for someone without an electronics background, if they wanted to make comparatively simple devices (e.g. a grad student in a non-ECE field wanting to create data loggers for experiments monitoring climate change variables), the system we describe would be invaluable, and enable this to happen at a high confidence without having to take hundreds of hours of coursework."
>
>
> **Clarification Response to questions for authors:**
>
> **"Question A: You say that the LLMs are instruction fine-tuned. What kind of instructions did you use and how?"**
> *Response:* We will clarify this description in the text.  We use GPT4 and Claude-V1 in this work, which are instruction-tuned models (i.e. they are pretrained on datasets of instructions, to understand user intent and accomplish the task the user describes).  It is not currently possible to fine-tune these models, and all experiments reported in this work are completed using in-context learning with the ~2k token prompt.
>
> **"Question B: Are there any procedures for what happens when the LLM suggests a part that is not available?"**
> *Response:* This is a great question.  When the LLM adds a part that is not available, we make every attempt to source it -- for example, the antique IN-12 Nixie Tube (and it's associated K155ID1 driver IC) were procured through dealers in legacy hardware.  An example we didn't include in the paper (but that is included in the repository, for transparency) is an open-generation device that's a magnetic compass that uses a magnetometer combined with a neopixel ring to create a navigational compass.  This device consistently generated with a HMC5883L magnetometer, which was a very popular magnetometer in the open source community that recently reached it's end-of-life.  We tried to produre it, but all the versions that were shipped ended up being unlicenced clone devices (e.g. the QMC5883L) that are not entirely compatible with standard libraries.  This complicates the evaluation (e.g. the code would require modification; but might have worked originally), so for this paper we simply didn't include it.  Ultimately this ends up being analogous to code generation tasks that generate code for libraries that are not only depricated, but also unavailable.  Our proposed solution is described breifly in the discussion (L440-445), where we call on building an open source simulator with an open part library to facilitate automated evaluation, but (also) to constrain design generation to a subset of available components.  While this is useful for end-of-life parts, you can imagine that this type of constraint-based generation might be particularly useful in a few years where a system like Words2Wires might be attached to a pick-and-place machine populated with a library of common parts -- so that you could ask the system to design a device for you and have it made moments later.  In such a case you would need to constrain generation to a subset of parts, and find ways to work with what you have (e.g. substituting components; combining two standard resistor values in series or parallel to make a desired value; etc.).
>
> **"Question C: You said that the devices were synthesized by a domain expert that had to correct some common errors. You also state that the LLM's capabilities are comparable to a student from an undergraduate course in microelectronics. Does the LLM reach undergraduate capabilities with or without the corrections? I am still to get a better understanding of the true capabilities of the LLMs."**
> *Response:* This is a good question, and we will clarify this in the text.  The text (L494-L498) comparing model performance to undergraduate performance is: "For context, being able to successfully design all the devices in the MICRO25 benchmark would be equivalent to the performance of a particularly strong undergraduate student after having taken a first course in microcontroller design at our institution.".  For clarity, this means the best model's performance as-is (e.g. uncorrected) on MICRO25 (GPT4, 96%) is equivalent to a strong (95%+ percentile) undergraduate at our institution.  Only the 6 case studies in Experiment 3, where the model is used as an assistant (instead of a PASS@1 generator) had errors corrected.
>
> **"Question D: Can you provide the prompts that you used to design the six real world use cases?"**
> *Response:* Yes!  These (and indeed all prompts used for all work in the paper) are provided in the Github repository with a script for replicating them.
>
> *Below are the progression of prompts used to create an ever-more-complex Emoji keyboard.  The model's generations were generally fine up until `keyboard-emoji5`, when the added requirement of playing a musical tune was added -- the model just generated a scaffold (e.g. void playMusic() // add music code here).  Since this was being used as a copilot-style assistant, we just fed that back into the model and asked it to populate that music code, but still count that as an error/omission in the original generation.*
> ```
> [keyboard-emoji1] create a keyboard that plugs in as a USB device, but instead of a full keyboard it has only a small number of buttons. The keyboard should only have buttons for 9 popular emojis.  There should be an LED that's on all the time.
>
> [keyboard-emoji2] create a keyboard that plugs in as a USB device, but instead of a full keyboard it has only a small number of buttons. The keyboard should only have buttons for 9 popular emojis, expressed as ASCII characters.  Some should be short, some can be long.  There should be an LED that's on all the time.
>
> [keyboard-emoji3] create a keyboard that plugs in as a USB device, but instead of a full keyboard it has only a small number of buttons. The keyboard should only have buttons for 9 popular emojis, expressed as ASCII characters.  One emoji should be a heart.  One other emoji should be a long string containing at least 5 characters.  There should be an LED that's on all the time, but blinks off for 500 milliseconds when a button is pressed.
>
> [keyboard-emoji4] create a keyboard that plugs in as a USB device, but instead of a full keyboard it has only a small number of buttons. The keyboard should only have buttons for 9 popular emojis, expressed as ASCII characters.  One emoji should be a heart.  One other emoji should be a long string containing at least 5 characters.  There should be an LED that's on all the time, but blinks off for 500 milliseconds when a button is pressed.
>
> [keyboard-emoji5] create a keyboard that plugs in as a USB device, but instead of a full keyboard it has only a small number of buttons. The keyboard should only have buttons for 9 popular emojis, expressed as ASCII characters.  One emoji should be a heart.  One other emoji should be a long string containing at least 5 characters.  There should be an LED that's on all the time, but blinks off for 500 milliseconds when a button is pressed.  There should also be a buzzer, that plays very brief music (i.e. no more than a few seconds) that is of the same affect as the emoji being pressed -- for example, loving for the heart emoji, happy for a happy emoji, sad music for a sad emoji, and so forth.
>
> [keyboard-emoji6] create a keyboard that plugs in as a USB device, but instead of a full keyboard it has only a small number of buttons. The keyboard should only have buttons for 9 popular emojis, expressed as ASCII characters, not unicode.  One emoji should be a heart.  There should be an LED that's on all the time, but blinks off for 500 milliseconds when a button is pressed.  There should also be a buzzer, that plays a brief tune that is of the same affect as the emoji being pressed -- for example, a love song for the heart emoji, a happy song for a happy emoji, sad music for a sad emoji, and so forth.
>
> [keyboard-emoji6a] create a keyboard that plugs in as a USB device, but instead of a full keyboard it has only a small number of buttons. The keyboard should only have buttons for 9 popular emojis, expressed as ASCII characters, not unicode.  One emoji should be a heart.  There should be an LED that's on all the time, but blinks off for 500 milliseconds when a button is pressed.  There should also be a piezo buzzer, that plays a brief tune that is of the same affect as the emoji being pressed -- for example, a love song for the heart emoji, a happy song for a happy emoji, sad music for a sad emoji, and so forth.
> ```
>
> **Remarks on reproducibility:**
>
> This is a work of open source science, and every effort has been made to make the resulting research products easy to use and well-documented for reproducibility.  The code, benchmarks, libraries, and results currently sit in a Github repository awaiting being made public.  Runscripts allow users to regenerate the results in about 12 hours (at current API speeds).  Unlike other tasks that require evaluation by domain experts (e.g. medical tasks, where doctors might disagree that a given symptom is individative of one diagnosis versus another), in our tasks the evaluations are objective -- i.e. the LED either blinks, or it doesn't; the servo motor either oscillates back and forth between 45 and 135 degrees, or it doesn't.  Papers that require (a) domain expertise, or (b) API access are still considered highly replicable (the PIs advisors used to have to travel across the world 20 years ago to access specific Linguistic Data Consortium (LDC) corpora to replicate others work).  Where possible and applicable, the results of this work are augmented with additional information (e.g. screenshots of the simulator, that augment the JSON netlists produced by the model) for usability and to quickly aid in verification.
>
> The reproducibility score suggests this work is not reproducible, but we believe this to be in error as every effort has been made to make this work  reflective of *highly reproducible* open source science.
>
> **Missing references:**
>
> Thank you for suggesting the additional reference (Thakur et al.) on Verilog code generation, and we will include this in our related work section.  But, it's worth noting that this work (published by IEEE and available online in June 2023 -- i.e. ACL-defined contemporaneous work): (a) Introduces a benchmark of 17 tasks (vs our 25 in Micro25, not including Pins100 or our 6 case studies), (b) has much less complex tasks than our benchmark (e.g. 8-bit adder vs Conway's Game of Life), and (c) being Verilog, is essentially a highly-constrained code-generation task evaluable entirely with unit tests, versus our open-ended hardware generation task that necessitates physical construction and expert evaluation.  Given that the work we report here is at a greater scale, scope, difficulty, and includes an expert-evaluation, and given the above concerns, this paper merits substantially higher scores than a hard reject (2 soundness / 3 excitement).

---

### Meta-Review · Area_Chair_qzuh · 2023-09-17

**Recommendation:** 3

**Metareview:**

As the AC for the paper, I commend the authors and reviewers for an extremely engaged discussion! Thank you to the authors for raising discussion points about quality reviewing practices, and to the reviewers for being receptive and open to these points. My metareview space is limited but I will summarize to the best of my ability.

In general, reviewers found this work to sufficiently support its major claims, although additional support was needed for some minor points. Reviewers found the paper to have strong merits (e.g., an innovative direction) but also key weaknesses (e.g., concerns regarding the evaluation) such that it could benefit from another round of revisions.

**Summary of Reviewer Feedback and Discussion:**
- **Reviewer P9mH** liked the framework provided by the authors, felt that the evaluation metrics and experiments were well-defined and thorough, and thought that the data could be a useful contribution to the community. However, they felt that the paper did not adequately distinguish itself from work relevant to code design for electronic circuits, and that an analysis of time savings from using LLMs could benefit the paper. They felt that the utility of the Micro25 dataset may be limited, requested details regarding the use of LLMs, and noted that reproducibility may be challenging since evaluating the work requires human expertise, in addition to asking clarifying questions. The authors responded that they did not discuss code generation applied to electronic circuits because their work is among the first in this subfield, and that their reliance on human evaluation is a strength. They noted that they are committed to releasing code, benchmarks, library, and experimental results, and that analyzing time savings is beyond the scope of their paper although an interesting direction for future work. They also provided detailed clarifying responses to the other questions.  In their response, Reviewer P9mH thanked the reviewers and clarified that it is important to include relevant papers to the extent possible to help situate the work in the broader research context.
- **Reviewer bhLT** thought the work was novel and appreciated the strong empirical results and new benchmarks. They noted that the inclusion of case studies validated the paper's claims and provided insights into practical applications, and they felt that the analysis was comprehensive and the work promised broad benefits to the NLP community. However, they felt that potential for end-to-end automation and scalability may be limited, were concerned by the lack of comparison with other potential approaches, and wished the paper had explored why LLMs were superior or necessary for this task. They also raised some clarifying questions. The authors responded that the design of high-level descriptions does not require domain expertise and the devices used in the paper reflect devices throughout the field, and that all coding assistants require human intervention. They explained that they did not compare to other approaches since their task was novel. Reviewer bhLT thanekd their authors and asked several follow-up questions, which the authors answered.
- **Reviewer 8oFB** liked that the paper presented benchmarks on a novel task, but felt that the high performance benchmarks may suggest that the task itself was trivial or that the data was not representative of the task's true difficulty. They felt that evaluation details were unclear and requested more information about the human expert, and they felt that the pass@k metric was inappropriate and that the merits for the case study were unclear, along with asking some clarifying questions. The authors disagreed that strong performance suggested task triviality, and noted that experimental and evaluation details were provided throughout the paper and appendix. They provided additional details regarding the human expert, and answered the clarifying questions.  Reviewer 8oFb thanked the authors and elaborated on their concerns, and the authors addressed these concerns.
- **Reviewer zUFP** appreciated the evaluation in a domain unfamiliar to many NLP researchers, and felt that the experimental setups were nicely selected. However, they found the evaluation to be superficial and had difficulty distinguishing the work from work towards automatic code generation. They also asked some clarifying questions. The authors responded that they used the standard evaluation metric for related tasks and that they included an error analysis in the appendix. They clarified the differences between their task and code generation tasks, and responded to the other questions. Reviewer zUFP noted that while the evaluation followed standard practices, they do not agree that it followed best practices, and noted that it would be nice to ultimately see what the task and results tell them about the language model, with NLP-focused insights (e.g., how the results relate to NLP tasks or language model capability).

---

### Decision · Program_Chairs · 2023-10-07

**Decision:**

Accept-Findings

**Comment:**

As the AC for the paper, I commend the authors and reviewers for an extremely engaged discussion! Thank you to the authors for raising discussion points about quality reviewing practices, and to the reviewers for being receptive and open to these points. My metareview space is limited but I will summarize to the best of my ability.

In general, reviewers found this work to sufficiently support its major claims, although additional support was needed for some minor points. Reviewers found the paper to have strong merits (e.g., an innovative direction) but also key weaknesses (e.g., concerns regarding the evaluation) such that it could benefit from another round of revisions.

**Summary of Reviewer Feedback and Discussion:**
- **Reviewer P9mH** liked the framework provided by the authors, felt that the evaluation metrics and experiments were well-defined and thorough, and thought that the data could be a useful contribution to the community. However, they felt that the paper did not adequately distinguish itself from work relevant to code design for electronic circuits, and that an analysis of time savings from using LLMs could benefit the paper. They felt that the utility of the Micro25 dataset may be limited, requested details regarding the use of LLMs, and noted that reproducibility may be challenging since evaluating the work requires human expertise, in addition to asking clarifying questions. The authors responded that they did not discuss code generation applied to electronic circuits because their work is among the first in this subfield, and that their reliance on human evaluation is a strength. They noted that they are committed to releasing code, benchmarks, library, and experimental results, and that analyzing time savings is beyond the scope of their paper although an interesting direction for future work. They also provided detailed clarifying responses to the other questions.  In their response, Reviewer P9mH thanked the reviewers and clarified that it is important to include relevant papers to the extent possible to help situate the work in the broader research context.
- **Reviewer bhLT** thought the work was novel and appreciated the strong empirical results and new benchmarks. They noted that the inclusion of case studies validated the paper's claims and provided insights into practical applications, and they felt that the analysis was comprehensive and the work promised broad benefits to the NLP community. However, they felt that potential for end-to-end automation and scalability may be limited, were concerned by the lack of comparison with other potential approaches, and wished the paper had explored why LLMs were superior or necessary for this task. They also raised some clarifying questions. The authors responded that the design of high-level descriptions does not require domain expertise and the devices used in the paper reflect devices throughout the field, and that all coding assistants require human intervention. They explained that they did not compare to other approaches since their task was novel. Reviewer bhLT thanekd their authors and asked several follow-up questions, which the authors answered.
- **Reviewer 8oFB** liked that the paper presented benchmarks on a novel task, but felt that the high performance benchmarks may suggest that the task itself was trivial or that the data was not representative of the task's true difficulty. They felt that evaluation details were unclear and requested more information about the human expert, and they felt that the pass@k metric was inappropriate and that the merits for the case study were unclear, along with asking some clarifying questions. The authors disagreed that strong performance suggested task triviality, and noted that experimental and evaluation details were provided throughout the paper and appendix. They provided additional details regarding the human expert, and answered the clarifying questions.  Reviewer 8oFb thanked the authors and elaborated on their concerns, and the authors addressed these concerns.
- **Reviewer zUFP** appreciated the evaluation in a domain unfamiliar to many NLP researchers, and felt that the experimental setups were nicely selected. However, they found the evaluation to be superficial and had difficulty distinguishing the work from work towards automatic code generation. They also asked some clarifying questions. The authors responded that they used the standard evaluation metric for related tasks and that they included an error analysis in the appendix. They clarified the differences between their task and code generation tasks, and responded to the other questions. Reviewer zUFP noted that while the evaluation followed standard practices, they do not agree that it followed best practices, and noted that it would be nice to ultimately see what the task and results tell them about the language model, with NLP-focused insights (e.g., how the results relate to NLP tasks or language model capability).